# FADE: Enabling Large-Scale Federated Adversarial Training on Resource-Constrained Edge Devices

## Abstract

Federated adversarial training can effectively complement adversarial robustness into the privacy-preserving federated learning systems. However, the high demand for memory capacity and computing power makes large-scale federated adversarial training infeasible on resource-constrained edge devices. Few previous studies in federated adversarial training have tried to tackle both memory and computational constraints at the same time. In this paper, we propose a new framework named Federated Adversarial Decoupled Learning (FADE) to enable AT on resource-constrained edge devices. FADE decouples the entire model into small modules to fit into the resource budget of each edge device respectively, and each device only needs to perform AT on a single module in each communication round. We also propose an auxiliary weight decay to alleviate objective inconsistency and achieve better accuracy-robustness balance in FADE. FADE offers a theoretical guarantee for convergence and adversarial robustness, and our experimental results show that FADE can significantly reduce the consumption of memory and computing power while maintaining accuracy and robustness.

## 1 Introduction

As a privacy-preserving distributed learning paradigm, Federated Learning (FL) makes a meaningful step toward the practice of secure and trustworthy artificial intelligence (Konečný et al., 2015; 2016; McMahan et al., 2017; Kairouz et al., 2019). In contrast to traditional centralized training, FL pushes the training to edge devices (clients), and client models are locally trained and uploaded to the server for aggregation. Since no private data is shared with other clients or the server, FL substantially improves the data privacy during the training process.

While FL can preserve the privacy of the participants, other threats can still impact the reliability of the machine learning model running on the FL system. One of such threats is adversarial samples, which aim to cause misclassifications of the model by adding imperceptible noise into the input data (Szegedy et al., 2013; Goodfellow et al., 2014). Previous research has shown that performing adversarial training (AT) on a large model is an effective method to attain robustness against adversarial samples while maintaining high accuracy on clean samples (Liu et al., 2020). However, large-scale AT also puts high demand for both memory capacity and computing power, which is affordable for some edge devices with limited resources, such as mobile phones and IoT devices, in FL scenarios (Kairouz et al., 2019; Li et al., 2020; Wong et al., 2020; Zizzo et al., 2020; Hong et al., 2021). Table 1 shows that strong robustness of the whole FL system cannot be attained by allowing only a small portion (e.g., 20%) of the clients to perform AT. Therefore, enabling resource-constrained edge devices (which usually contribute to the majority of the participants in cross-device FL (Kairouz et al., 2019)) to perform AT is necessary for achieving strong robustness in FL.

Some previous works have tried to tackle client-wise systematic heterogeneity in FL (Li et al., 2018; Lu et al., 2020; Wang et al., 2020b; Xie et al., 2019). The most common method to deal with the slow devices is to allow them performing less epochs of local training than the others (Li et al., 2018; Wang et al., 2020b). While this method can reduce the computational costs on the slow devices, the memory capacity limitation on edge devices has not been well discussed in these works.

Table 1: Results of partial federated adversarial training with 100 clients. "20% AT + 80% ST" means that 20% clients perform AT while 80% clients perform standard training (ST).

| Training Scheme | FMNIST (CNN-7) | | CIFAR-10 (VGG-11) | |
|---|---|---|---|---|
| | Natural Acc. | Adversarial Acc. | Natural Acc. | Adversarial Acc. |
| 100% AT + 0% ST | 78.39% | 66.93% | 64.73% | 33.27% |
| 20% AT + 80% ST | 83.83% | 48.61% | 74.77% | 19.22% |

To tackle both memory capacity and computational constraints, recent studies propose a novel training scheme named Decoupled Greedy Learning (DGL) which decouples the entire neural network into several small modules and trains each module separately (Belilovsky et al., 2019; Wang et al., 2021). DGL can be naturally deployed in FL since the training of decoupled modules can be parallelized on different computing nodes (Belilovsky et al., 2020). However, vanilla DGL only supports a unique model partition on all computing nodes, which cannot fit into different resource budgets of different clients in heterogeneous FL. Additionally, no previous studies have explored whether DGL can be combined with AT to confer joint adversarial robustness to the entire model. It is not trivial to achieve joint robustness of the entire model when applying AT in DGL, since modules are trained separately in DGL with different locally supervised losses.

In this paper, we propose **F**ederated **A**dversarial **DE**coupled Learning (FADE), which is the first adversarial decoupled learning scheme for heterogeneous FL. Our main contributions are:

1. We propose a more flexible decoupled learning scheme for heterogeneous Federated Learning, which allows different model partitions on devices with different resource budgets. We give a theoretical guarantee for the convergence of our Federated Decoupled Learning.

2. We propose **F**ederated **A**dversarial **DE**coupled Learning (FADE) to attain theoretically guaranteed joint adversarial robustness of the entire model. Our experimental results show that FADE can significantly reduce the memory and computational requirements while maintaining the natural accuracy and adversarial robustness as joint training.

3. We analyze the trade-off between objective consistency (natural accuracy) and adversarial robustness (adversarial accuracy) in FADE, and we propose an effective method to achieve a better accuracy-robustness balance point with the weight decay on auxiliary models.

## 2 PRELIMINARY

**Federated Learning (FL)**   In FL, different clients collaboratively train a shared global model $\boldsymbol{w}$ with locally stored data (McMahan et al., 2017). The objective of FL can be formulated as:

$$\min_{\boldsymbol{w}} \quad L(\boldsymbol{w}) = \frac{1}{\sum_i |\mathbb{D}_i|} \sum_{k=1}^{N} \sum_{(\boldsymbol{x},y) \in \mathbb{D}_k} l(\boldsymbol{x}, y; \boldsymbol{w}) = \sum_{k=1}^{N} q_k L_k(\boldsymbol{w}), \tag{1}$$

$$\text{where} \quad L_k(\boldsymbol{w}) = \frac{1}{|\mathbb{D}_k|} \sum_{(\boldsymbol{x},y) \in \mathbb{D}_k} l(\boldsymbol{x}, y; \boldsymbol{w}) = \mathbb{E}_{(\boldsymbol{x},y) \sim \mathbb{D}_k} \left[ l(\boldsymbol{x}, y; \boldsymbol{w}) \right], \tag{2}$$

and $l$ is the task loss, e.g., cross-entropy loss for classification task. $\mathbb{D}_k$ is the dataset of client $k$ and its weight $q_k = |\mathbb{D}_k|/(\sum_i |\mathbb{D}_i|)$. To solve for the optimal solution of this objective, in each communication round, FL first samples a subset of clients $\mathbb{S}^{(t)}$ to perform local training. These clients initialize their models with the global model $\boldsymbol{w}_k^{(t,0)} = \boldsymbol{w}^{(t)}$, and then run $\tau$ iterations of local SGD. After all these clients complete training in this round, their models are uploaded and averaged to become the new global model (McMahan et al., 2017). We summarize this procedure as follows:

$$\boldsymbol{w}_k^{(t+1)} = \boldsymbol{w}^{(t)} - \eta_t \sum_{j=0}^{\tau-1} \nabla L_k(\boldsymbol{w}_k^{(t,j)}), \tag{3}$$

$$\boldsymbol{w}^{(t+1)} = \frac{1}{\sum_{i \in \mathbb{S}^{(t)}} q_i} \sum_{k \in \mathbb{S}^{(t)}} q_k \boldsymbol{w}_k^{(t+1)}, \tag{4}$$

where $\boldsymbol{w}_k^{(t,j)}$ is the local model of client $k$ at the $j$-th iteration of round $t$.

**Adversarial Training (AT)** The goal of AT is to achieve robustness against small perturbation in the inputs. We define $(\epsilon, c)$-robustness as follows:

**Definition 1.** *We say a model $\boldsymbol{w}$ is $(\epsilon, c)$-robust in a loss function $l$ at input $\boldsymbol{x}$ if*

$$\forall \boldsymbol{\delta} \in \{\boldsymbol{\delta} : \|\boldsymbol{\delta}\|_p \leq \epsilon\}, l(\boldsymbol{x} + \boldsymbol{\delta}, y; \boldsymbol{w}) - l(\boldsymbol{x}, y; \boldsymbol{w}) \leq c, \tag{5}$$

*where $\|\cdot\|_p$ is the $\ell_p$ norm of a vector[1], and $\epsilon$ is the perturbation tolerance.*

AT trains a model with adversarial samples to achieve adversarial robustness, which can be formulated as a min-max problem (Goodfellow et al., 2014; Madry et al., 2017):

$$\min_{\boldsymbol{w}} \max_{\boldsymbol{\delta} : \|\boldsymbol{\delta}\|_p \leq \epsilon} l(\boldsymbol{x} + \boldsymbol{\delta}, y; \boldsymbol{w}). \tag{6}$$

To solve Eq. 6, people usually alternatively solve the inner maximization and the outer minimization. While solving the inner maximization, Projected Gradient Descent (PGD) is shown to introduce the strongest robustness in AT (Madry et al., 2017; Wong et al., 2020; Wang et al., 2021).

**Decoupled Greedy Learning (DGL)** The key idea of DGL is to decouple the entire model into several non-overlapping small modules. By introducing a locally supervised loss to each module, we can load and train each module independently without accessing the other parts of the entire model (Belilovsky et al., 2019; 2020). This enables devices with small memory to train large models.

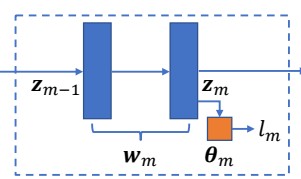

Figure 1: An illustration of the module $m$.

As shown in Fig. 1, each module $m$ usually contains one or multiple adjacent layers $\boldsymbol{w}_m$ of the backbone neural network, together with a small auxiliary model $\boldsymbol{\theta}_m$ that provides locally supervised loss. We denote $\boldsymbol{\Theta}_m = (\boldsymbol{w}_m, \boldsymbol{\theta}_m)$ to be all the parameters in module $m$. Module $m$ accepts the features $\boldsymbol{z}_{m-1}$ from the previous module as the input, and it outputs features $\boldsymbol{z}_m = f_m(\boldsymbol{z}_{m-1}; \boldsymbol{w}_m)$ for the following modules, as well as a locally supervised loss $l_m(\boldsymbol{z}_{m-1}, y; \boldsymbol{\Theta}_m)$. At epoch $t$, the averaged locally supervised loss $L_m^{(t)}$ will be used for training this module:

$$L_m^{(t)}(\boldsymbol{\Theta}_m^{(t)}) = \mathbb{E}_{(\boldsymbol{z}_{m-1}^{(t)}, y)} \left[ l_m(\boldsymbol{z}_{m-1}^{(t)}, y; \boldsymbol{\Theta}_m^{(t)}) \right]. \tag{7}$$

Different from joint training, the input of one module can be various in different epochs in DGL since we may keep updating the previous modules during training. Thus we use $\boldsymbol{z}_{m-1}^{(t)}$ to denote the inputs of module $m$ in epoch $t$, and only the input of the first module $\boldsymbol{z}_0^{(t)} = \boldsymbol{x}$ is invariant.

For each module $m \in \{1, 2, \cdots, M\}$ in the entire model, we define the loss function of its auxiliary model as $\tilde{l}_m(\boldsymbol{z}_m, y; \boldsymbol{\theta}_m) = l_m(\boldsymbol{z}_{m-1}, y; \boldsymbol{\Theta}_m)$, and the loss function of its following layers in the backbone network as $\tilde{l}'_m(\boldsymbol{z}_m, y; \boldsymbol{w}_{m+1}, \cdots, \boldsymbol{w}_M) = l(\boldsymbol{x}, y; \boldsymbol{w}_1, \cdots, \boldsymbol{w}_M)$. Without specifying, we will omit all parameters ($\boldsymbol{w}_m, \boldsymbol{\theta}_m$ and $\boldsymbol{\Theta}_m$) in the following sections for notation simplicity.

## 3 FEDERATED ADVERSARIAL DECOUPLED LEARNING

In this section, we present our method, Federated Adversarial Decoupled Learning (FADE), which aims at enabling all clients with different computing resources to participate in adversarial training. We first introduce Federated Decoupled Learning (FDL) with flexible model partitions for heterogeneous FL in Section 3.1, and we also give a convergence analysis for it. In Section 3.2, we integrate AT into FDL to achieve joint adversarial robustness of the entire model, and we give a theoretical guarantee for its robustness. In Section 3.3, we discuss the objective inconsistency in FDL and propose an effective method to achieve a better accuracy-robustness balance point.

### 3.1 FEDERATED DECOUPLED LEARNING

In cross-device FL, the main participants are usually small edge devices who have limited hardware resources and may not be able to afford large-scale AT that requires large memory and high computing power (Li et al., 2018; Kairouz et al., 2019; Li et al., 2020; Wang et al., 2020b). A solution

---

[1]For simplicity, without specifying $p$, we use $\|\cdot\|$ for $\ell_2$ norm. Our conclusions in the following sections can be extended to any $\ell_p$ norm with the equivalence of vector norms.

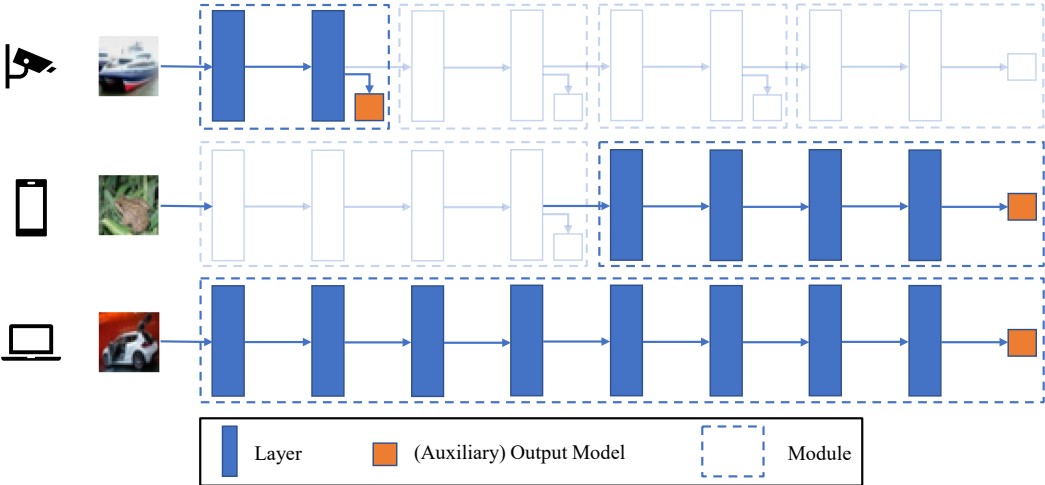

Figure 2: A framework of Federated Decoupled Learning. In contrast to the original unique partition (Belilovsky et al., 2020), we allow different model partitions among devices according to their resource budgets. In each communication round, each device randomly selects one module (highlighted) for training, and then the updates of each layer will be averaged respectively.

to tackle the resource constraints on edge devices is to deploy DGL in FL, where each device only needs to load and train a single module instead of the entire model in each communication round. However, the vanilla DGL only supports a unique model partition on all the devices (Belilovsky et al., 2020). Considering the systematic heterogeneity, we would prefer various model partitions to fit into different resource budgets of different clients. A device with limited resources (such as a IoT device) can train small modules of the entire model, while a device with more resources (such as a mobile phone or a computer) can train larger modules or even the entire model.

Accordingly, we propose our Federated Decoupled Learning (FDL) framework as shown in Fig. 2. We denote the set of all modules on client $k$ as $\mathbb{M}_k$, while $\mathbb{M}_i \neq \mathbb{M}_j$ if client $i$ is using a different model partition from client $j$. Here, we consider the update and aggregation rule for each layer $n$ with parameter $\boldsymbol{\omega}_n$ in the model, since one single layer is the "atom" in FDL and cannot be further decoupled. We use $m_k(n)$ to denote the module on client $k$ that contains this layer, and we define $L_{n,k} = L_{m_k(n),k}$ as the locally supervised loss for training this layer. In each communication round $t$, each client $k$ randomly samples a module $m_k^t$ from $\mathbb{M}_k$ for training (Eq. 8). After the local training, the updates of each layer $n$ will be averaged over clients in $\mathbb{S}_n^{(t)}$ respectively, where $\mathbb{S}_n^{(t)}$ is the set of clients whose trained module $m_k^t$ contains layer $n$ in this round(Eq. 9).

$$\boldsymbol{\omega}_{n,k}^{(t+1)} = \begin{cases} \boldsymbol{\omega}_n^{(t)} - \eta_t \sum_{j=0}^{\tau-1} \nabla_{\boldsymbol{\omega}_n} L_{n,k}^{(t)}, & \text{if } n \in m_k^t; \\ \boldsymbol{\omega}_n^{(t)}, & \text{elsewhere.} \end{cases} \qquad (8)$$

$$\boldsymbol{\omega}_n^{(t+1)} = \frac{1}{\sum_{i \in \mathbb{S}_n^{(t)}} q_i} \sum_{k \in \mathbb{S}_n^{(t)}} q_k \boldsymbol{\omega}_{n,k}^{(t+1)}, \quad \text{where } \mathbb{S}_n^{(t)} = \{k \in \mathbb{S}^{(t)} : n \in m_k^t\}. \qquad (9)$$

Theorem 1 guarantees the convergence of FDL, while the full version with proof is in Appendix A.

**Theorem 1.** *Under some common assumptions, for any layer $n$ in the entire model, its locally supervised loss $L_n = \sum_k q_k L_{n,k}$ can converge in Federated Decoupled Learning:*

$$\lim_{T \to \infty} \inf_{t \leq T} \mathbb{E} \left\| \nabla_{\boldsymbol{\omega}_n} L_n \right\|^2 = 0. \qquad (10)$$

Theorem 1 can guarantee the convergence of locally supervised loss $L_n$. However, because of the existence of the objective inconsistency $\|\nabla L - \nabla L_n\| \geq 0$, we cannot guarantee the convergence of the joint loss $L$ with this result. We discuss the objective inconsistency in Section 3.3, and we show how we can reduce this gap such that we can make the joint loss gradient $\nabla L$ smaller when the locally supervised loss $L_n$ converges.

## 3.2 Adversarial Decoupled Learning

Adversarial decoupled learning can achieve local robustness of each module by performing AT in each module $m$ separately on their own locally supervised loss:

$$\min_{\boldsymbol{\Theta}_m} \max_{\boldsymbol{\delta}_{m-1}} \quad l_m(\boldsymbol{z}_{m-1} + \boldsymbol{\delta}_{m-1}, y; \boldsymbol{\Theta}_m), \quad \text{subject to } \|\boldsymbol{\delta}_{m-1}\| \leq \epsilon_{m-1}. \tag{11}$$

However, there are two concerns that have not been addressed in adversarial decoupled learning:

1. Since different modules are trained with different locally supervised losses, can local robustness of each module guarantee the joint robustness of the entire (backbone) model?

2. When applying AT on a module $m$, what value of the perturbation tolerance $\epsilon_{m-1}$ should we use to ensure the joint robustness of the entire model?

Theorem 2 reveals the relationship between the local robustness of each module and the joint robustness of the entire model, and it gives a lower bound of the perturbation tolerance $\epsilon_{m-1}$ for each module $m$ to sufficiently guarantee the joint robustness. Theorem 2 is proved in Appendix B.1.

**Theorem 2.** *Assume that $\tilde{l}_m(\boldsymbol{z}_m, y)$ is $\mu_m$-strongly convex in $\boldsymbol{z}_m$ for each module $m$. If each module $m \leq M$ has local $(\epsilon_{m-1}, c_m)$-robustness in $l_m(\boldsymbol{z}_{m-1}, y)$, and*

$$\forall m \leq M, \quad \epsilon_m \geq \frac{g_m}{\mu_m} + \sqrt{\frac{2c_m}{\mu_m} + \frac{g_m^2}{\mu_m^2}}, \quad \text{where } g_m = \|\nabla_{\boldsymbol{z}_m} \tilde{l}_m(\boldsymbol{z}_m, y)\|, \tag{12}$$

*then we can guarantee that the entire model has a joint $(\epsilon_0, c_M)$-robustness in $l(\boldsymbol{x}, y)$.*

**Remark 1.** In Theorem 2, we assume that the loss function of the auxiliary model $\tilde{l}_m(\boldsymbol{z}_m, y)$ is strongly convex in its input $\boldsymbol{z}_m$. This assumption is realistic since the auxiliary model is usually a very simple model, e.g., only a linear layer followed by cross-entropy loss. We also theoretically analyze the sufficiency of a simple auxiliary model in Section 3.3 (See Remark 2).

Theorem 2 shows that a larger $\mu_m$ and a smaller $g_m$ will lead to a stronger joint robustness of the entire model, since the lower bound of $\epsilon_m$ becomes smaller for ensuring the joint robustness. In Section 3.3, we further discuss how we can control these two parameters to attain better accuracy-robustness balance with the weight decay on the auxiliary model.

## 3.3 Objective Inconsistency and Accuracy-Robustness Trade-off

As we mentioned in Section 3.1, there exists objective inconsistency between the module and the entire model because the module is trained with locally supervised loss $l_m$ instead of the joint loss $l$ (Wang et al., 2021). The objective inconsistency in FDL is defined by the difference between the gradients of the locally supervised loss ($\nabla_{\boldsymbol{w}_m} l_m$) and the joint loss ($\nabla_{\boldsymbol{w}_m} l$). The existence of this inconsistency makes the optimal parameters that minimize the locally supervised loss $l_m$ does not necessarily minimize the joint loss $l$. Furthermore, the objective inconsistency can enlarge heterogeneity among clients and hinder the convergence of FL (Li et al., 2019; Wang et al., 2020b), thus it is important to alleviate the objective inconsistency in FDL to improve its performance.

Theorem 3 shows a non-trivial relationship between adversarial robustness and objective inconsistency: strong joint adversarial robustness also implies small objective inconsistency in FDL. We prove Theorem 3 in Appendix B.2.

**Theorem 3.** *Assume that $\tilde{l}_m(\boldsymbol{z}_m, y)$ and $\tilde{l}'_m(\boldsymbol{z}_m, y)$ are $\beta_m, \beta'_m$-smooth in $\boldsymbol{z}_m$ for a module $m$. If there exist $c_m$, $c'_m$, and $r \geq \sqrt{2\frac{c_m + c'_m}{\beta_m + \beta'_m}}$, such that the auxiliary model has $(r, c_m)$-robustness in $\tilde{l}_m(\boldsymbol{z}_m, y)$, and the backbone network has $(r, c'_m)$-robustness in $\tilde{l}'_m(\boldsymbol{z}_m, y)$, then we have:*

$$\|\nabla_{\boldsymbol{w}_m} l - \nabla_{\boldsymbol{w}_m} l_m\| \leq \left\|\frac{\partial \boldsymbol{z}_m}{\partial \boldsymbol{w}_m}\right\| \sqrt{2(c_m + c'_m)(\beta_m + \beta'_m)}. \tag{13}$$

Theorem 3 suggests that we can alleviate the objective inconsistency by reducing $\beta_m, \beta'_m, c_m$ and $c'_m$ (Regularizing $\|\partial \boldsymbol{z}_m / \partial \boldsymbol{w}_m\|$ usually requires second derivative, which introduces high memory

---

**Algorithm 1** FADE: Federated Adversarial Decoupled Learning

---

1: Initialize $\boldsymbol{w}^{(0)}$ and $\boldsymbol{\theta}_m^{(0)}$ for each module $m$.
2: **for** $t = 1, 2, \cdots, T$ **do**
3:     Randomly sample a group of clients $\mathbb{S}^{(t)}$ for training.
4:     **for** each client $k \in \mathbb{S}^{(t)}$ in parallel **do**
5:         Randomly select a module $m_k^t$ that will be trained in this round.
6:         Request the current global model $\boldsymbol{w}^{(t)}$ and the auxiliary model $\boldsymbol{\theta}_{m_k^t}^{(t)}$ from the server.
7:         Generate input features $\boldsymbol{z}_{m_k^t-1}^{(t)}$ for all data $\boldsymbol{x} \in \mathbb{D}_k$.
8:         Perform AT in module $m_k^t$ with $l_{m_k^t}^{\text{FADE}}$ in Eq. 14 for $\tau$ iterations, and get $\boldsymbol{\Theta}_{m_k^t,k}^{(t+1)}$.
9:         Upload $\boldsymbol{\Theta}_{m_k^t,k}^{(t+1)}$ to the server.
10:     **end for**
11:     The server aggregates $\boldsymbol{\omega}_k^{(t+1)}$ to get $\boldsymbol{\omega}^{(t+1)}$ according to Eq. 9 for each $\boldsymbol{\omega}$.
12: **end for**

---

and computational overhead, so we do not consider it here). Notice that $c_m'$ is small given the joint robustness of the backbone network, which is guaranteed by adversarial decoupled learning in Theorem 2. Furthermore, Moosavi-Dezfooli et al. (2019) shows that adversarial robustness also implies a smoother loss function. Therefore, the joint robustness also leads to a small $\beta_m'$.

Accordingly, with adversarial decoupled learning, we only need to reduce $\beta_m$ and $c_m$ to alleviate the objective inconsistency. We notice that both $\beta_m$ and $c_m$ are only related to the auxiliary model, and we show in Appendix B.3 that we can reduce them by adding a large weight decay on the auxiliary model $\boldsymbol{\theta}_m$ when the auxiliary model is simple (e.g., only a single linear layer).

**Remark 2.** It is noteworthy that we do not use any conditions on the difference between $\tilde{l}_m'$ and $\tilde{l}_m$ in both Theorem 2 and 3. This implies that the auxiliary model is not required to perform as well as the joint backbone model. Thus, a simple auxiliary model is sufficient to achieve high joint robustness and low objective inconsistency in adversarial decoupled learning.

Based on all analysis above, we propose Federated Adversarial Decoupled Learning (FADE), where we replace the original loss function $l_m$ in Eq. 7 by the following adversarial loss with weight decay:

$$l_m^{\text{FADE}}(\boldsymbol{z}_{m-1}^{(t)}, y; \boldsymbol{w}_m^{(t)}, \boldsymbol{\theta}_m^{(t)}) = \max_{\boldsymbol{\delta}_{m-1}^{(t)}} \left[ l_m(\boldsymbol{z}_{m-1}^{(t)} + \boldsymbol{\delta}_{m-1}^{(t)}, y; \boldsymbol{w}_m^{(t)}, \boldsymbol{\theta}_m^{(t)}) \right] + \lambda_m \|\boldsymbol{\theta}_m^{(t)}\|^2, \qquad (14)$$

where $\lambda_m$ is the hyperparameter that control the weight decay on the auxiliary model $\boldsymbol{\theta}_m$. Our framework is summarized in Algorithm 1.

**Trade-off Between Joint Accuracy and Joint Robustness.** As we discussed in Section 3.2 and this section, four parameters ($\mu_m, g_m, c_m$ and $\beta_m$) that are only related to the auxiliary model $\boldsymbol{\theta}_m$ can influence the joint robustness and objective consistency. We can see in Appendix B.3 that applying a larger $\lambda_m$ can decrease all of them. Smaller $c_m$ and $\beta_m$ can alleviate the objective inconsistency to increase the joint accuracy, and smaller $g_m$ can improve the joint robustness. However, smaller $\mu_m$ will lead to weaker robustness by increasing the lower bound of $\epsilon_m$. Therefore, there exists accuracy-robustness trade-off when we apply the weight decay, and the value of $\lambda_m$ plays an important role in balancing the joint accuracy and the joint robustness.

## 4 EXPERIMENTAL RESULTS

### 4.1 EXPERIMENT SETTINGS

We conduct our experiments on two datasets, FMNIST (Xiao et al., 2017) and CIFAR-10 (Krizhevsky et al., 2009). To simulate the statistical heterogeneity in FL, we partition the whole dataset into $N = 100$ clients with the same Non-IID data partition as Shah et al. (2021), where 80% data of each client is from only two classes while 20% is from the other eight classes. We sample $C = 30$ clients for local training in each communication round. We conduct two groups of

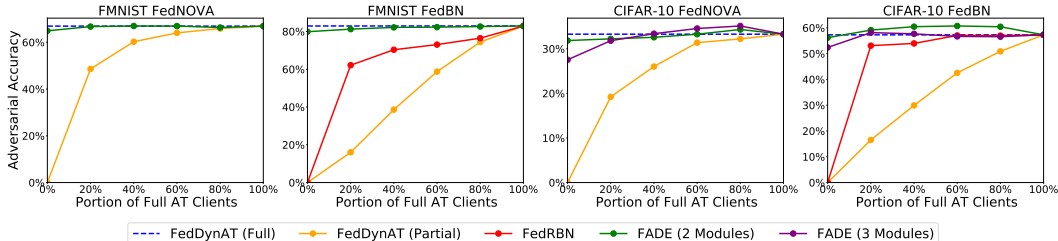

Figure 3: Minimum memory and computational requirements of baselines and FADE. The results are shown as the percentage of the resource requirement of full FedDynAT with PGD-10 AT.

Figure 4: Adversarial accuracy when training with different portions of resource-sufficient clients.

experiments with two state-of-the-art FL optimizers respectively: FedNOVA (Wang et al., 2020b) for global FL and FedBN (Li et al., 2021b) for personalized FL. Notice that the results in global FL and personalized FL are not comparable since they assume different test set partitions. We combine FADE with different FL optimizers to show the generalization of our method.

For AT settings, following Moosavi-Dezfooli et al. (2019) and Zizzo et al. (2020), we use $l_\infty$ norm to bound the perturbation and use PGD-10 to generate adversarial samples for training and test. The perturbation tolerance at input $z_0 = x$ is set to be $\epsilon_0 = 0.15$ with PGD step size $\alpha_0 = 0.03$ for FMNIST. For CIFAR-10, we set $\epsilon_0 = 8/255$ and $\alpha_0 = 2/255$.

For FMNIST, we use a 7-layer CNN (CNN-7) with five convolutional layers and two fully connected layers. We adopt a model partition with 2 modules for CNN-7. For CIFAR-10, we use VGG-11 (Simonyan & Zisserman, 2014) as the model. We adopt two different model partitions for VGG-11, with 2 modules and 3 modules respectively. See Appendix C for more details.

In the following sections, we will compare our method FADE with three baselines. Full FedDynAT (Shah et al., 2021) represents the ideal performance of federated adversarial training when all the clients are able to perform AT on the entire model. While FedDynAT with $100\%$ AT is not feasible under our limitation that only a small portion of clients can afford AT on the entire model, we adopt partial FedDynAT where clients with insufficient resources only perform standard training (ST). Another baseline FedRBN (Hong et al., 2021) also allows resource-constrained devices performing ST only, and the robustness will be propagated by transferring the batch-normalization statistics from the clients who can afford AT to the clients who only perform ST.

## 4.2 Resource Requirements

We measure the minimum resource requirements of FADE and all baselines on resource-constrained devices. We use the number of loaded parameters as the metric of memory, and we use FLOPs as the metric of computation. For partial FedDynAT and FedRBN, the minimum memory requirement is the number of parameters in the entire model since they always load the entire model for training, and the computing power requirement is the FLOPs of ST on the entire model since the resource-constrained devices only perform ST. For FADE, the minimum memory requirement is the number of parameters in the largest module, while the computing power requirement is the mean of FLOPs for PGD-10 AT across all modules. The results are shown in Fig. 3.

We can see that FADE can reduce the memory requirement by more than $40\%$ on both CNN-7 and VGG-11, while FADE with 2 modules and 3 modules can reduce the computation by $50\%$ and $67\%$ respectively. Although partial FedDynAT and FedRBN can largely reduce the amount

Table 2: The natural accuracy (clean samples) and adversarial accuracy (adversarial samples) on FMNIST. Results are reported in the mean and the standard deviation over 3 random seeds.

| Training Scheme | FedNOVA | | FedBN | |
| --- | --- | --- | --- | --- |
| | Natural Acc. | Adversarial Acc. | Natural Acc. | Adversarial Acc. |
| FedDynAT (100% AT) | $78.39 \pm 0.65\%$ | $66.93 \pm 0.87\%$ | $89.85 \pm 0.41\%$ | $82.92 \pm 0.66\%$ |
| FedDynAT (20% AT) | $83.83 \pm 0.32\%$ | $48.61 \pm 0.87\%$ | $91.94 \pm 0.07\%$ | $16.02 \pm 1.03\%$ |
| FedRBN | n/a | n/a | $90.35 \pm 1.51\%$ | $62.14 \pm 6.45\%$ |
| FADE (2 Modules) | $78.74 \pm 1.09\%$ | $66.72 \pm 2.09\%$ | $89.43 \pm 0.52\%$ | $81.24 \pm 0.94\%$ |

Table 3: The natural accuracy (clean samples) and adversarial accuracy (adversarial samples) on CIFAR-10. Results are reported in the mean and the standard deviation over 3 random seeds.

| Training Scheme | FedNOVA | | FedBN | |
| --- | --- | --- | --- | --- |
| | Natural Acc. | Adversarial Acc. | Natural Acc. | Adversarial Acc. |
| FedDynAT (100% AT) | $64.73 \pm 1.63\%$ | $33.27 \pm 0.46\%$ | $81.71 \pm 0.14\%$ | $57.28 \pm 1.23\%$ |
| FedDynAT (20% AT) | $74.77 \pm 1.68\%$ | $19.22 \pm 2.16\%$ | $87.12 \pm 0.25\%$ | $16.51 \pm 1.64\%$ |
| FedRBN | n/a | n/a | $86.80 \pm 0.31\%$ | $53.08 \pm 1.03\%$ |
| FADE (2 Modules) | $65.42 \pm 0.42\%$ | $32.22 \pm 0.43\%$ | $81.05 \pm 0.56\%$ | $59.12 \pm 0.63\%$ |
| FADE (3 Modules) | $64.72 \pm 0.68\%$ | $31.81 \pm 0.35\%$ | $77.46 \pm 0.67\%$ | $58.14 \pm 0.85\%$ |
| FADE (Mixing) | $66.06 \pm 1.09\%$ | $32.28 \pm 0.49\%$ | $78.23 \pm 0.35\%$ | $58.80 \pm 0.66\%$ |

of computation, they are far less efficient than they appear to be when training a large model that exceeds the memory limit, since they need to repeatedly fetch and load small parts of the entire model from the cloud or the external storage during each forward and backward propagation. And we will also see in the following experiments that neither of them can maintain the adversarial robustness, while FADE can still achieve the same level of robustness as full FedDynAT.

### 4.3 PERFORMANCE OF FADE

We first compare our method with three baselines under the limitation that only 20% clients can afford AT on the entire model, while the other 80% clients can only afford Standard Training (ST) on the entire model or AT on a module. The natural and adversarial accuracy in FMNIST and CIFAR-10 is shown in Table 2 and 3 respectively. While neither partial FedDynAT nor FedRBN can maintain the robustness under the resource constraint, FADE consistently outperforms other baselines and achieves almost the same or even higher accuracy and robustness comparing to full FedDynAT (the constraint-free case). In addition, we mix clients with one module (joint training), clients with two modules and clients with three modules in a ratio of 2:3:5 as the setting "FADE (Mixing)". We can see that FADE still attains high accuracy and robustness in this case, which verifies the compatibility of our flexible FDL framework.

We also conducted experiments with different proportions of resource-sufficient clients who perform joint AT on the entire model, and the adversarial accuracy is shown in Fig. 4. Even in the worst case that none of the clients have enough resources to complete AT on the entire model, FADE can achieve robustness comparable to full FedDynAT. And with only 40% resource-sufficient clients, FADE can already attain the same robustness as full FedDynAT in all our experiments, while the other baselines still have significant robustness gaps from full FedDynAT.

### 4.4 THE INFLUENCE OF WEIGHT DECAY ON THE AUXILIARY MODEL

As we suggested in Section 3.3, the auxiliary model weight decay hyperparameter $\lambda_m$ acts as an important role that balances natural accuracy and adversarial robustness. To show the influence of this hyperparameter, we conduct experiments in FADE (2 Modules) with different $\lambda_m$ between 0.0001 and 0.1, and we plot the natural and adversarial accuracy in Fig. 5.

We can observe that in all our settings the natural accuracy increases first as we increase $\lambda_m$, and then goes down quickly. The growing part can be explained by our theory in Section 3.3 that the larger auxiliary weight decay can alleviate the objective inconsistency and improve the performance. However, when we adopt a too large weight decay, the weight decay will drive the model away from optimum and lead to a performance drop, which is also commonly observed in joint training process.

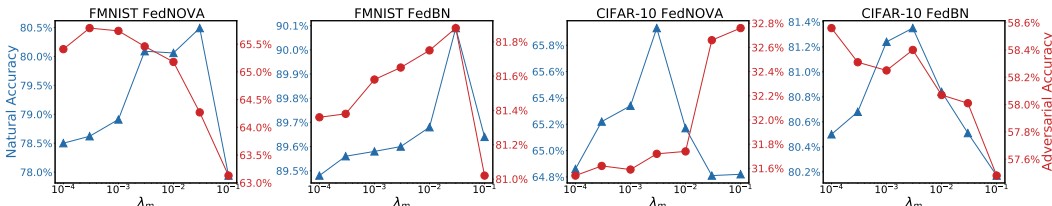

Figure 5: Natural (blue lines with triangle markers) and adversarial (red lines with circle markers) accuracy with different auxiliary weight decay hyperparameter $\lambda_m$.

For the adversarial accuracy, the effects of $\lambda_m$ become more complicated, since larger $\lambda_m$ can decrease both $g_m$ and $\mu_m$, which affect the robustness in opposite ways. An increasing adversarial accuracy suggests that the effect of $g_m$ dominates, while a decreasing one suggests that the effect of $\mu_m$ dominates. Similarly to the natural accuracy, we could observe that the adversarial accuracy usually grows first before going down, which implies that the effect of $g_m$ is usually stronger when $\lambda_m$ is small. And considering the increasing natural accuracy, adopting a moderately large $\lambda_m$ usually attains a better overall performance on clean and adversarial samples.

## 5 RELATED WORKS

**Federated Learning** Client-wise heterogeneity is one of the challenges that hinders the practice of Federated Learning (FL). Many studies have tried to overcome the statistical heterogeneity in data (Karimireddy et al., 2019; Liang et al., 2019; Tang et al., 2022; Wang et al., 2020a) and the systematic heterogeneity in hardware (Li et al., 2021a; 2018; Wang et al., 2020b). Beyond the heterogeneity, FL is also vulnerable in several kinds of attack, such as model poisoning attack (Bhagoji et al., 2019; Sun et al., 2021) and adversarial sample attack (Zizzo et al., 2020; Shah et al., 2021). In this paper, we mainly focus on the adversarial sample attack and deal with the challenge in federated adversarial training under client-wise heterogeneity (Hong et al., 2021).

**Adversarial Training** AT is well known for its high demand for computing resources (Wong et al., 2020). Several fast AT algorithms have been proposed to reduce the computation in AT (Shafahi et al., 2019; Zhang et al., 2019), such as replacing PGD with FGSM (Andriushchenko & Flammarion, 2020; Wong et al., 2020) or using other regularization methods for robustness (Moosavi-Dezfooli et al., 2019; Qin et al., 2019). FADE can be easily combined with these fast AT algorithms to further reduce the computing cost, which we leave as a future work. In addition, AT will decrease the model performance on clean samples, and thus a larger model is usually required to maintain the same natural accuracy (Liu et al., 2020). This makes AT also memory-demanding.

**Decoupled Greedy Learning** As deeper and deeper neural networks are used for better performance, the low efficiency of end-to-end (joint) training is exposed because it hinders the model parallelization and requires large memory for model parameters and intermediate results (Belilovsky et al., 2020; Hettinger et al., 2017). As an alternative, Decoupled Greedy Learning (DGL) is proposed, which decouples the whole neural network into several modules and trains them separately without gradient dependency (Belilovsky et al., 2019; 2020; Marquez et al., 2018; Wang et al., 2021). As a more flexible DGL framework, FADE fits better in heterogeneous FL while offering guarantees in convergence as well as joint adversarial robustness.

## 6 CONCLUSIONS

In this paper, we proposed Federated Adversarial Decoupled Learning (FADE), a novel framework to reduce the memory and computing power requirements for resource-constrained edge devices in large-scale federated adversarial training. Our theory guarantees the convergence and joint adversarial robustness of FADE, and we develop an effective regularizer to reduce the objective inconsistency in FADE based on the theory. Our experimental results show that FADE can significantly reduce both memory and computing power consumption on small edge devices, while maintaining almost the same accuracy as the joint federated adversarial training on both clean and adversarial samples.

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

# A    Convergence Analysis of Federated Decoupled Learning

## A.1    Preliminary

In this section, we analyze the convergence property of Federated Decoupled Learning (FDL). Since FDL partitions the entire model with layers as the smallest unit, we only need to prove the convergence of each layer. We use $\boldsymbol{\omega}_n$ to denote all the parameters in layer $n$, and $m_k(n)$ to denote the module that contains layer $n$ on client $k$. We define the parameters other than $\boldsymbol{\omega}_n$ in module $m_k(n)$ as $\boldsymbol{\Omega}_{n,k}$. We also denote the input feature of layer $n$ as $\boldsymbol{z}_{n-1,k} = \boldsymbol{z}_{m_k(n)}$ on client $k$. we define the locally supervised loss of layer $n$ on client $k$ as:

$$l_{n,k}^{(t,j)}(\boldsymbol{z}_{n-1}^{(t)}, y; \boldsymbol{\omega}_n^{(t,j)}) = l_{m_k(n)}(\boldsymbol{z}_{m_k(n)-1}^{(t)}, y; \boldsymbol{\omega}_n^{(t,j)}, \boldsymbol{\Omega}_{n,k}^{(t,j)}), \tag{15}$$

where $l_{n,k}^{(t,j)}$ changes every iteration because of the update of $\boldsymbol{\Omega}_{n,k}^{(t,j)}$. For simplicity, from now on we abridge $(\boldsymbol{z}, y)$ as $\boldsymbol{z}$. We let $\boldsymbol{z}_{n-1,k}^{(t)}$ follow the distribution with probability density $p_{n-1,k}^{(t)}(\mathbf{z})$ at the $j$-th iteration of communication round $t$, and we define its converged density as $p_{n-1,k}^*(\mathbf{z})$ with converged previous layers and $\boldsymbol{\Omega}_{n,k}^*$ (Belilovsky et al., 2020). With these notations, we define

$$L_{n,k}^{(t)}(\boldsymbol{\omega}_n^{(t)}) = \mathbb{E}_{\boldsymbol{z}_{n-1,k}^{(t)} \sim p_{n-1,k}^{(t)}} \left[ \frac{1}{\tau} \sum_{j=0}^{\tau-1} l_{n,k}^{(t,j)}(\boldsymbol{z}_{n-1,k}^{(t)}; \boldsymbol{\omega}_n^{(t)}) \right]; \tag{16}$$

$$L_n^{(t)}(\boldsymbol{\omega}_n^{(t)}) = \sum_{k=1}^N q_k L_{n,k}^{(t)}(\boldsymbol{\omega}_n^{(t)}); \tag{17}$$

$$L_{n,k}(\boldsymbol{\omega}_n^{(t)}) = \mathbb{E}_{\boldsymbol{z}_{n-1,k} \sim p_{n-1,k}^*} \left[ l_{n,k}^*(\boldsymbol{z}_{n-1,k}; \boldsymbol{\omega}_n^{(t)}) \right]$$

$$= \mathbb{E}_{\boldsymbol{z}_{m_k(n)-1,k} \sim p_{m_k(n)-1,k}^*} \left[ l_{m_k(n)}(\boldsymbol{z}_{m_k(n)-1,k}; \boldsymbol{\omega}_n^{(t)}, \boldsymbol{\Omega}_{n,k}^*) \right]; \tag{18}$$

$$L_n(\boldsymbol{\omega}_n^{(t)}) = \sum_{k=1}^N q_k L_{n,k}(\boldsymbol{\omega}_n^{(t)}); \tag{19}$$

Following Belilovsky et al. (2020), we use the distance between the current density and the converged density below for our analysis:

$$\rho_{n-1}^{(t)} \triangleq \sum_{k=1}^N q_k \int \left| p_{n-1,k}^{(t)}(\mathbf{z}) - p_{n-1,k}^*(\mathbf{z}) \right| d\mathbf{z}, \tag{20}$$

And we also define the following gap between $l_{n,k}^{(t)}$ and $l_{n,k}^*$:

$$\xi_n^{(t)} \triangleq \sum_{k=1}^N \sum_{j=0}^{\tau-1} \frac{q_k}{\tau} \left\| \mathbb{E}_{\boldsymbol{z}_{n-1,k} \sim p_{n-1}^*} \left[ \nabla l_{n,k}^{(t,j)}(\boldsymbol{z}_{n-1,k}; \boldsymbol{\omega}_n^{(t)}) - \nabla l_{n,k}^*(\boldsymbol{z}_{n-1,k}; \boldsymbol{\omega}_n^{(t)}) \right] \right\|^2 \tag{21}$$

We will discuss the convergence of $L_n(\boldsymbol{\omega}_n)$ for each layer $n$. Without specifying, all the gradients ($\nabla L$ or $\nabla l$) in the following analysis are with respect to $\boldsymbol{\omega}_n$. Following Belilovsky et al. (2020) and Wang et al. (2020b), we make the common assumptions below.

**Assumption 1** ($\mathcal{L}$-smoothness (Belilovsky et al., 2020; Wang et al., 2020b))**.** *$L_n$ is differentiable with respect to $\boldsymbol{\omega}_n^{(t)}$ and its gradient is $\mathcal{L}_n$-Lipschitz for all $t$. Similarly, $L_{n,k}^{(t)}$ is differentiable with respect to $\boldsymbol{\omega}_{n,k}^{(t)}$ and its gradient is $\widetilde{\mathcal{L}}_n$-Lipschitz for all $t$ and $k$.*

**Assumption 2** (Robbins-Monro conditions (Belilovsky et al., 2020))**.** *The learning rates satisfy $\sum_{t=0}^\infty \eta_t = \infty$ yet $\sum_{t=0}^\infty \eta_t^2 < \infty$.*

**Assumption 3** (Finite variance (Belilovsky et al., 2020; Wang et al., 2020b))**.** *There exists some positive constant $G$ such that $\forall t, j$ and $\forall k$, $\mathbb{E}_{\boldsymbol{z}_{n-1,k}^{(t,j)} \sim p_{n-1,k}^{(t,j)}} \left[ \left\| \nabla l_{n,k}^{(t,j)}(\boldsymbol{z}_{n-1,k}^{(t,j)}; \boldsymbol{\omega}_n) \right\|^2 \right] \le G$ and*

*$\mathbb{E}_{\boldsymbol{z}_{n-1,k} \sim p_{n-1}^*} \left[ \left\| \nabla l_{n,k}^*(\boldsymbol{z}_{n-1,k}; \boldsymbol{\omega}_n) \right\|^2 \right] \le G$ at any $\boldsymbol{\omega}_n$.*

**Assumption 4** (Bounded Dissimilarity (Wang et al., 2020b)). *There exist constants $\beta^2 \geq 1, \kappa^2 \geq 0$ such that $\forall t$ and $\boldsymbol{\omega}_n$, $\sum_{k=1}^N q_k \left\| \nabla L_{n,k}^{(t)}(\boldsymbol{\omega}_n) \right\|^2 \leq \beta^2 \left\| \sum_{k=1}^N q_k \nabla L_{n,k}^{(t)}(\boldsymbol{\omega}_n) \right\|^2 + \kappa^2$.*

**Assumption 5** (Convergence of the previous modules and $\boldsymbol{\Omega}_n$ (Belilovsky et al., 2020)). *We assume that $\sum_{t=0}^\infty \rho_{n-1}^{(t)} < \infty$ and $\sum_{t=0}^\infty \xi_n^{(t)} < \infty$.*

### A.2 PROOF OF THEOREM 1

With all above assumptions, we get the following theorem that guarantees the convergence of Federated Decoupled Learning.

**Theorem 1.** *Under Assumption 1- 5, Federated Decoupled Learning converges as follows:*

$$
\inf_{t \leq T} \mathbb{E} \left[ \left\| \nabla L_n \left( \boldsymbol{\omega}_n^{(t)} \right) \right\|^2 \right]
$$
$$
\leq \mathcal{O} \left( \frac{1}{\sum_{t=0}^T \eta_t} \right) + \mathcal{O} \left( \frac{\sum_{t=0}^T \rho_n^{(t)} \eta_t}{\sum_{t=0}^T \eta_t} \right) + \mathcal{O} \left( \frac{\sum_{t=0}^T \xi_n^{(t)} \eta_t}{\sum_{t=0}^T \eta_t} \right) + \mathcal{O} \left( \frac{\sum_{t=0}^T \eta_t^2}{\sum_{t=0}^T \eta_t} \right). \tag{22}
$$

*Proof.* We consider the SGD scheme in Eq. 9 with learning rate $\{\eta_t\}_t$:

$$
\boldsymbol{\omega}_n^{(t+1)} = \boldsymbol{\omega}_n^{(t)} - \eta_t \frac{\sum_{k \in \mathbb{S}_n^{(t)}} q_k \boldsymbol{h}_{n,k}^{(t)}}{\sum_{k \in \mathbb{S}_n^{(t)}} q_k},
$$

where $\mathbb{S}_n^{(t)} = \mathbb{S}_{\boldsymbol{\omega}_n}^{(t)}$ which is defined in Eq. 9. And $\boldsymbol{h}_{n,k}^{(t)}$ is defined as

$$
\boldsymbol{h}_{n,k}^{(t)} = \sum_{j=0}^{\tau-1} \nabla l_{n,k}^{(t,j)} (\boldsymbol{z}_{n-1,k}^{(t,j)}; \boldsymbol{\omega}_{n,k}^{(t,j)}). \tag{23}
$$

According to the Lipschitz-smooth assumption for the global objective function $L_n$, it follows that

$$
\mathbb{E} \left[ L_n \left( \boldsymbol{\omega}_n^{(t+1)} \right) \right] - L_n \left( \boldsymbol{\omega}_n^{(t)} \right)
$$
$$
\leq -\eta_t \underbrace{\mathbb{E} \left[ \left\langle \nabla L_n \left( \boldsymbol{\omega}_n^{(t)} \right), \frac{\sum_{k \in \mathbb{S}_n^{(t)}} q_k \boldsymbol{h}_{n,k}^{(t)}}{\sum_{k \in \mathbb{S}_n^{(t)}} q_k} \right\rangle \right]}_{T_1} + \frac{\eta_t^2 \mathcal{L}_n}{2} \underbrace{\mathbb{E} \left[ \left\| \frac{\sum_{k \in \mathbb{S}_n^{(t)}} q_k \boldsymbol{h}_{n,k}^{(t)}}{\sum_{k \in \mathbb{S}_n^{(t)}} q_k} \right\|^2 \right]}_{T_2}. \tag{24}
$$

Similar to the proof in (Wang et al., 2020b), to bound the $T_1$ in Inequality 24, we should notice that

$$
T_1 = \mathbb{E} \left[ \left\langle \nabla L_n \left( \boldsymbol{\omega}_n^{(t)} \right), \frac{\sum_{k \in \mathbb{S}_n^{(t)}} q_k \boldsymbol{h}_{n,k}^{(t)}}{\sum_{k \in \mathbb{S}_n^{(t)}} q_k} \right\rangle \right]
$$
$$
= \left\langle \nabla L_n \left( \boldsymbol{\omega}_n^{(t)} \right), \sum_{k=1}^N q_k \mathbb{E} \boldsymbol{h}_{n,k}^{(t)} \right\rangle
$$
$$
= \frac{1}{2} \left\| \nabla L_n \left( \boldsymbol{\omega}_n^{(t)} \right) \right\|^2 + \frac{1}{2} \left\| \sum_{k=1}^N q_k \mathbb{E} \boldsymbol{h}_{n,k}^{(t)} \right\|^2 - \frac{1}{2} \left\| \nabla L_n \left( \boldsymbol{\omega}_n^{(t)} \right) - \sum_{k=1}^N q_k \mathbb{E} \boldsymbol{h}_{n,k}^{(t)} \right\|^2 \tag{25}
$$
$$
\geq \frac{1}{2} \left\| \nabla L_n \left( \boldsymbol{\omega}_n^{(t)} \right) \right\|^2 - \frac{1}{2} \left\| \nabla L_n \left( \boldsymbol{\omega}_n^{(t)} \right) - \sum_{k=1}^N q_k \mathbb{E} \boldsymbol{h}_{n,k}^{(t)} \right\|^2
$$
$$
\geq \frac{1}{2} \left\| \nabla L_n \left( \boldsymbol{\omega}_n^{(t)} \right) \right\|^2 - \left\| \nabla L_n^{(t)} \left( \boldsymbol{\omega}_n^{(t)} \right) - \sum_{k=1}^N q_k \mathbb{E} \boldsymbol{h}_{n,k}^{(t)} \right\|^2 - \left\| \nabla L_n \left( \boldsymbol{\omega}_n^{(t)} \right) - \nabla L_n^{(t)}(\boldsymbol{\omega}_n^{(t)}) \right\|^2
$$
$$
\tag{26}
$$

$$\geq \frac{1}{2}\left\|\nabla L_n\left(\boldsymbol{\omega}_n^{(t)}\right)\right\|^2 - \sum_{k=1}^{N} q_k \left\|\mathbb{E}\boldsymbol{h}_{n,k}^{(t)} - \nabla L_{n,k}^{(t)}(\boldsymbol{\omega}_n^{(t)})\right\|^2 - \left\|\nabla L_n\left(\boldsymbol{\omega}_n^{(t)}\right) - \nabla L_n^{(t)}(\boldsymbol{\omega}_n^{(t)})\right\|^2.$$
(27)

Eq. 25 uses the fact: $2\langle a, b\rangle = \|a\|^2 + \|b\|^2 - \|a-b\|^2$, and Inequality 26 uses the fact: $\|a+b\|^2 \leq 2\|a\|^2 + 2\|b\|^2$. Inequality 27 uses $L_n^{(t)} = \sum_{k=1}^{N} q_k L_{n,k}^{(t)}$ and Jenson's inequality $\|\sum_{i=1}^{m} b_i a_i\|^2 \leq \sum_{i=1}^{m} b_i \|a_i\|^2$.

Based on the proof of Lemma 3.2 in Belilovsky et al. (2020), we have

$$\left\|\nabla L_n\left(\boldsymbol{\omega}_n^{(t)}\right) - \nabla L_n^{(t)}\left(\boldsymbol{\omega}_n^{(t)}\right)\right\|^2$$

$$= \left\|\sum_k q_k \mathbb{E}_{\boldsymbol{z}\sim p_{n-1,k}^*}\left[\nabla l_{n,k}^*(\boldsymbol{z};\boldsymbol{\omega}_n^{(t)})\right] - \sum_k q_k \mathbb{E}_{\boldsymbol{z}\sim p_{n-1,k}^{(t)}}\left[\frac{1}{\tau}\sum_j \nabla l_{n,k}^{(t,j)}(\boldsymbol{z};\boldsymbol{\omega}_n^{(t)})\right]\right\|^2$$

$$= \left\|\frac{1}{\tau}\sum_k q_k \sum_j \left(\mathbb{E}_{\boldsymbol{z}\sim p_{n-1,k}^*}\left[\nabla l_{n,k}^*(\boldsymbol{z};\boldsymbol{\omega}_n^{(t)})\right] - \mathbb{E}_{\boldsymbol{z}\sim p_{n-1,k}^{(t)}}\left[\nabla l_{n,k}^{(t,j)}(\boldsymbol{z};\boldsymbol{\omega}_n^{(t)})\right]\right)\right\|^2$$

$$\leq \frac{1}{\tau}\sum_k q_k \sum_j \left\|\mathbb{E}_{\boldsymbol{z}\sim p_{n-1,k}^*}\left[\nabla l_{n,k}^*(\boldsymbol{z};\boldsymbol{\omega}_n^{(t)})\right] - \mathbb{E}_{\boldsymbol{z}\sim p_{n-1,k}^{(t)}}\left[\nabla l_{n,k}^{(t,j)}(\boldsymbol{z};\boldsymbol{\omega}_n^{(t)})\right]\right\|^2$$

$$\leq \frac{2}{\tau}\sum_k q_k \sum_j \left\|\int \nabla l_{n,k}^{(t,j)}(\boldsymbol{z};\boldsymbol{\omega}_n^{(t)})p_{n-1,k}^{(t)}(\boldsymbol{z})d\boldsymbol{z} - \int \nabla l_{n,k}^{(t,j)}(\boldsymbol{z};\boldsymbol{\omega}_n^{(t)})p_{n-1,k}^*(\boldsymbol{z})d\boldsymbol{z}\right\|^2$$

$$+ \frac{2}{\tau}\sum_k q_k \sum_j \left\|\mathbb{E}_{\boldsymbol{z}\sim p_{n-1,k}^*}\left[\nabla l_{n,k}^{(t,j)}(\boldsymbol{z};\boldsymbol{\omega}_n^{(t)}) - \nabla l_{n,k}^*(\boldsymbol{z};\boldsymbol{\omega}_n^{(t)})\right]\right\|^2$$

$$\leq \frac{2}{\tau}\sum_k q_k \sum_j \left(\int \left\|\nabla l_{n,k}^{(t,j)}(\boldsymbol{z};\boldsymbol{\omega}_n^{(t)})\right\| \sqrt{|p_{n-1,k}^{(t)}(\boldsymbol{z}) - p_{n-1,k}^*(\boldsymbol{z})||p_{n-1,k}^{(t)}(\boldsymbol{z}) - p_{n-1,k}^*(\boldsymbol{z})|}d\boldsymbol{z}\right)^2$$

$$+ 2\xi_n^{(t)}$$

$$\leq \frac{2}{\tau}\sum_k q_k \sum_j \int \left\|\nabla l_{n,k}^{(t,j)}(\boldsymbol{z};\boldsymbol{\omega}_n^{(t)})\right\|^2 |p_{n-1,k}^{(t)}(\boldsymbol{z}) - p_{n-1}^*(\boldsymbol{z})|d\boldsymbol{z} \int |p_{n-1,k}^{(t)}(\boldsymbol{z}) - p_{n-1,k}^*(\boldsymbol{z})|d\boldsymbol{z}$$

$$+ 2\xi_n^{(t)}$$

$$\leq \frac{2}{\tau}\sum_k q_k \int |p_{n-1,k}^{(t)}(\boldsymbol{z}) - p_{n-1,k}^*(\boldsymbol{z})|d\boldsymbol{z} \sum_j \int \left\|\nabla l_{n,k}^{(t,j)}(\boldsymbol{z};\boldsymbol{\omega}_n^{(t)})\right\|^2 \left(p_{n-1,k}^{(t)}(\boldsymbol{z}) + p_{n-1,k}^*(\boldsymbol{z})\right)d\boldsymbol{z}$$

$$+ 2\xi_n^{(t)}$$

$$\leq 4G\rho_n^{(t)} + 2\xi_n^{(t)}$$
(28)

Hence, we have

$$T_1 \geq \frac{1}{2}\left\|\nabla L_n\left(\boldsymbol{\omega}_n^{(t)}\right)\right\|^2 - \sum_{k=1}^{N} q_k \left\|\mathbb{E}\boldsymbol{h}_{n,k}^{(t)} - \nabla L_{n,k}^{(t)}(\boldsymbol{\omega}_n^{(t)})\right\|^2 - 4G\rho_n^{(t)} - 2\xi_n^{(t)}.$$
(29)

Similar to the proof in Section C.3, we have the following bound for $T_2$:

$$T_2 \leq 2\left[\mathbb{E}\left\|\frac{\sum_{k\in\mathbb{S}_n^{(t)}} q_k \boldsymbol{h}_{n,k}^{(t)}}{\sum_{k\in\mathbb{S}_n^{(t)}} q_k} - \frac{\sum_{k\in\mathbb{S}_n^{(t)}} q_k \mathbb{E}\boldsymbol{h}_{n,k}^{(t)}}{\sum_{k\in\mathbb{S}_n^{(t)}} q_k}\right\|^2\right] + 2\left[\mathbb{E}\left\|\frac{\sum_{k\in\mathbb{S}_n^{(t)}} q_k \mathbb{E}\boldsymbol{h}_{n,k}^{(t)}}{\sum_{k\in\mathbb{S}_n^{(t)}} q_k}\right\|^2\right]$$

$$\leq 4\left[\mathbb{E}\left\|\frac{\sum_{k\in\mathbb{S}_n^{(t)}} q_k \boldsymbol{h}_{n,k}^{(t)}}{\sum_{k\in\mathbb{S}_n^{(t)}} q_k}\right\|^2 + \mathbb{E}\left\|\frac{\sum_{k\in\mathbb{S}_n^{(t)}} q_k \mathbb{E}\boldsymbol{h}_{n,k}^{(t)}}{\sum_{k\in\mathbb{S}_n^{(t)}} q_k}\right\|^2\right] + 2\left[\mathbb{E}\left\|\frac{\sum_{k\in\mathbb{S}_n^{(t)}} q_k \mathbb{E}\boldsymbol{h}_{n,k}^{(t)}}{\sum_{k\in\mathbb{S}_n^{(t)}} q_k}\right\|^2\right]$$

$$
\leq 4 \left[ \mathbb{E} \frac{\sum_{k \in \mathbb{S}_n^{(t)}} q_k \|\boldsymbol{h}_{n,k}^{(t)}\|^2}{\sum_{k \in \mathbb{S}_n^{(t)}} q_k} + \mathbb{E} \frac{\sum_{k \in \mathbb{S}_n^{(t)}} q_k \mathbb{E}\|\boldsymbol{h}_{n,k}^{(t)}\|^2}{\sum_{k \in \mathbb{S}_n^{(t)}} q_k} \right] + 2 \left[ \mathbb{E} \left\| \frac{\sum_{k \in \mathbb{S}_n^{(t)}} q_k \mathbb{E}\boldsymbol{h}_{n,k}^{(t)}}{\sum_{k \in \mathbb{S}_n^{(t)}} q_k} \right\|^2 \right]
$$

$$
\leq 8\tau^2 G + 2 \left[ \mathbb{E} \left\| \frac{\sum_{k \in \mathbb{S}_n^{(t)}} q_k \mathbb{E}\boldsymbol{h}_{n,k}^{(t)}}{\sum_{k \in \mathbb{S}_n^{(t)}} q_k} \right\|^2 \right] \tag{30}
$$

$$
\leq 8\tau^2 G + 6 \sum_{k=1}^N q_k \left\| \mathbb{E}\boldsymbol{h}_{n,k}^{(t)} - \nabla L_{n,k}^{(t)}(\boldsymbol{\omega}_n^{(t)}) \right\|^2
$$

$$
+ 6\|\nabla L_n^{(t)}(\boldsymbol{\omega}_n^{(t)})\|^2 + \frac{6}{S_t} \left( \beta^2 \|\nabla L_n^{(t)}(\boldsymbol{\omega}_n^{(t)})\|^2 + \kappa^2 \right), \tag{31}
$$

where $S_t = \left| \mathbb{S}_n^{(t)} \right|$. Inequality 30 is based on the Assumption 3 and the definition of $\boldsymbol{h}_{n,k}^{(t)}$, while Inequality 31 is from Lemma 5 in Wang et al. (2020b).

According to Assumption 3, for all $t, j, k, \boldsymbol{\omega}_n$, we have

$$
\mathbb{E}_{\boldsymbol{z} \sim p_{n-1,k}^{(t)}} \left[ \left\| \nabla l_{n,k}^{(t,j)}(\boldsymbol{z}; \boldsymbol{\omega}_n) - \mathbb{E}_{\boldsymbol{z} \sim p_{n-1,k}^{(t)}} \nabla l_{n,k}^{(t,j)}(\boldsymbol{z}; \boldsymbol{\omega}_n) \right\|^2 \right]
$$

$$
\leq \mathbb{E}_{\boldsymbol{z} \sim p_{n-1,k}^{(t)}} \left[ 2 \left\| \nabla l_{n,k}^{(t,j)}(\boldsymbol{z}; \boldsymbol{\omega}_n) \right\|^2 + 2\|\mathbb{E}_{\boldsymbol{z} \sim p_{n-1,k}^{t)}} \nabla l_{n,k}^{(t,j)}(\boldsymbol{z}; \boldsymbol{\omega}_n)\|^2 \right]
$$

$$
\leq 4\mathbb{E}_{\boldsymbol{z} \sim p_{n-1,k}^{(t)}} \left\| \nabla l_{n,k}^{(t,j)}(\boldsymbol{z}; \boldsymbol{\omega}_n) \right\|^2
$$

$$
\leq 4G. \tag{32}
$$

With the results in Inequality 32 and in Section C.5 of Wang et al. (2020b), we have the following bound

$$
\frac{1}{2} \sum_{k=1}^N q_k \left\| \mathbb{E}\boldsymbol{h}_{n,k}^{(t)} - \nabla L_{n,k}^{(t)}(\boldsymbol{\omega}_n^{(t)}) \right\|^2
$$

$$
\leq \frac{4\eta_t^2 \widetilde{\mathcal{L}}_n^2 G}{1 - D}(\tau^2 - 1) + \frac{D\beta^2}{2(1-D)} \left\| \nabla L_n^{(t)}(\boldsymbol{\omega}_n^{(t)}) \right\|^2 + \frac{D\kappa^2}{2(1-D)}, \tag{33}
$$

where $D = 4\eta_t^2 \widetilde{\mathcal{L}}_n^2 \tau(\tau - 1) < 1$. If $D \leq \frac{1}{12\beta^2+1}$, then it follows that $\frac{1}{1-D} \leq 1 + \frac{1}{12\beta^2} \leq 2$ and $\frac{3D\beta^2}{1-D} \leq \frac{1}{4}$. In this case, we can further simplify the inequality:

$$
6 \sum_{k=1}^N q_k \left\| \mathbb{E}\boldsymbol{h}_{n,k}^{(t)} - \nabla L_{n,k}^{(t)}(\boldsymbol{\omega}_n^{(t)}) \right\|^2
$$

$$
\leq 96\eta_t^2 \widetilde{\mathcal{L}}_n^2 G(\tau^2 - 1) + \frac{1}{2} \left\| \nabla L_n^{(t)}(\boldsymbol{\omega}_n^{(t)}) \right\|^2 + 48\eta_t^2 \widetilde{\mathcal{L}}_n^2 \kappa^2 \tau(\tau - 1)
$$

$$
\leq 96\eta_t^2 \widetilde{\mathcal{L}}_n^2 G(\tau^2 - 1) + \left\| \nabla L_n(\boldsymbol{\omega}_n^{(t)}) \right\|^2 + 4G\rho_n^{(t)} + 2\xi_n^{(t)} + 48\eta_t^2 \widetilde{\mathcal{L}}_n^2 \kappa^2 \tau(\tau - 1). \tag{34}
$$

Then we can bound $T_2$ as follows:

$$
T_2 \leq 8\tau^2 G + 6 \sum_{k=1}^N q_k \left\| \mathbb{E}\boldsymbol{h}_{n,k}^{(t)} - \nabla L_{n,k}^{(t)}(\boldsymbol{\omega}_n^{(t)}) \right\|^2
$$

$$
+ 6\|\nabla L_n^{(t)}(\boldsymbol{\omega}_n^{(t)})\|^2 + \frac{6}{S_t} \left( \beta^2 \|\nabla L_n^{(t)}(\boldsymbol{\omega}_n^{(t)})\|^2 + \kappa^2 \right)
$$

$$
\leq 8\tau^2 G + 96\eta_t^2 \widetilde{\mathcal{L}}_n^2 G(\tau^2 - 1) + \left\| \nabla L_n(\boldsymbol{\omega}_n^{(t)}) \right\|^2 + 4G\rho_n^{(t)} + 2\xi_n^{(t)} + 48\eta_t^2 \widetilde{\mathcal{L}}_n^2 \kappa^2 \tau(\tau - 1)
$$

$$
+ 6\|\nabla L_n(\boldsymbol{\omega}_n^{(t)})\|^2 + 48G\rho_n^{(t)} + 24\xi_n^{(t)}
$$

$$
+ \frac{6}{S_t} \left( \beta^2 \|\nabla L_n(\boldsymbol{\omega}_n^{(t)})\|^2 + 8\beta^2 G\rho_n^{(t)} + 4\beta^2 \xi_n^{(t)} + \kappa^2 \right), \tag{35}
$$

where Inequality 35 uses the difference bound in Inequality 28. Plugging Inequality 29 and Inequality 35 back into Inequality 24, and with $S_t \geq 1$, we have

$$
\mathbb{E}\left[L_n\left(\boldsymbol{\omega}_n^{(t+1)}\right)\right] - L_n\left(\boldsymbol{\omega}_n^{(t)}\right)
$$

$$
\leq -\frac{1}{2}\eta_t \left\|\nabla L_n\left(\boldsymbol{\omega}_n^{(t)}\right)\right\|^2 + 4\eta_t G\rho_n^{(t)} + 2\eta_t \xi_n^{(t)}
$$

$$
+ \eta_t\left[16\eta_t^2\widetilde{\mathcal{L}}_n^2 G(\tau^2-1) + \frac{1}{6}\left\|\nabla L_n(\boldsymbol{\omega}_n^{(t)})\right\|^2 + \frac{2}{3}G\rho_n^{(t)} + \frac{1}{3}\xi_n^{(t)} + 8\eta_t^2\widetilde{\mathcal{L}}_n^2\kappa^2\tau(\tau-1)\right]
$$

$$
+ \frac{\eta_t^2\mathcal{L}_n}{2}\left[8\tau^2 G^2 + 96\eta_t^2\widetilde{\mathcal{L}}_n^2 G(\tau^2-1) + \left\|\nabla L_n(\boldsymbol{\omega}_n^{(t)})\right\|^2 + 4G\rho_n^{(t)} + 2\xi_n^{(t)}\right.
$$

$$
+ 48\eta_t^2\widetilde{\mathcal{L}}_n^2\kappa^2\tau(\tau-1) + 6\|\nabla L_n(\boldsymbol{\omega}_n^{(t)})\|^2 + 48G\rho_n^{(t)} + 24\xi_n^{(t)}
$$

$$
\left. + 6(\beta^2\|\nabla L_n(\boldsymbol{\omega}_n^{(t)})\|^2 + 8\beta^2 G\rho_n^{(t)} + 4\beta^2\xi_n^{(t)} + \kappa^2)\right] \tag{36}
$$

$$
\leq -\left(\frac{5}{12}\eta_t - \frac{7\eta_t^2\mathcal{L}_n}{2} - 3\eta_t^2\mathcal{L}_n\beta^2\right)\left\|\nabla L_n\left(\boldsymbol{\omega}_n^{(t)}\right)\right\|^2
$$

$$
+ \eta_t\left[\frac{14}{3}G\rho_n^{(t)} + \frac{7}{3}\xi_n^{(t)} + 16\eta_t^2\widetilde{\mathcal{L}}_n^2 G(\tau^2-1) + 8\eta_t^2\widetilde{\mathcal{L}}_n^2\kappa^2\tau(\tau-1)\right]
$$

$$
+ \frac{\eta_t^2\mathcal{L}_n}{2}\left[8\tau^2 G^2 + 52G\rho_n^{(t)} + 26\xi_n^{(t)} + 96\eta_t^2\widetilde{\mathcal{L}}_n^2 G(\tau^2-1) + 48\eta_t^2\widetilde{\mathcal{L}}_n^2\kappa^2\tau(\tau-1)\right.
$$

$$
\left. + 48\beta^2 G\rho_n^{(t)} + 24\beta^2\xi_n^{(t)} + 6\kappa^2\right]. \tag{37}
$$

When we set $\eta_t \leq \min\{\frac{1}{(21+18\beta^2)\mathcal{L}_n}, 1\}$, we can get

$$
\frac{1}{4}\eta_t\|\nabla L_n\left(\boldsymbol{\omega}_n^{(t)}\right)\|^2
$$

$$
\leq L_n\left(\boldsymbol{\omega}_n^{(t)}\right) - \mathbb{E}\left[L_n\left(\boldsymbol{\omega}_n^{(t+1)}\right)\right]
$$

$$
+ \left(\frac{14}{3}G + 26\mathcal{L}_n G + 24\beta^2\mathcal{L}_n G\right)\eta_t\rho_n^{(t)} + \left(\frac{7}{3} + 13\mathcal{L}_n + 12\beta^2\mathcal{L}_n\right)\eta_t\xi_n^{(t)}
$$

$$
+ \left[16\widetilde{\mathcal{L}}_n^2 G(\tau^2-1) + 8\widetilde{\mathcal{L}}_n^2\kappa^2\tau(\tau-1) + 4\mathcal{L}_n\tau^2 G^2\right.
$$

$$
\left. + 48\widetilde{\mathcal{L}}_n^2\mathcal{L}_n G(\tau^2-1) + 24\widetilde{\mathcal{L}}_n^2\mathcal{L}_n\kappa^2\tau(\tau-1) + 3\mathcal{L}_n\kappa\right]\eta_t^2.
$$

Taking the expectation and averaging across all rounds, one can obtain

$$
\frac{1}{4}\sum_{t=0}^{T}\eta_t\left\|\nabla L_n\left(\boldsymbol{\omega}_n^{(t)}\right)\right\|^2
$$

$$
\leq \left[L_n\left(\boldsymbol{\omega}_n^{(0)}\right) - L_n\left(\boldsymbol{\omega}_n^{(T+1)}\right)\right] + A\sum_{t=0}^{T}\eta_t^2 + B\sum_{t=0}^{T}\eta_t\rho_n^{(t)} + C\sum_{t=0}^{T}\eta_t\xi_n^{(t)}
$$

$$
\leq L_n\left(\boldsymbol{\omega}_n^{(0)}\right) + A\sum_{t=0}^{T}\eta_t^2 + B\sum_{t=0}^{T}\eta_t\rho_n^{(t)} + C\sum_{t=0}^{T}\eta_t\xi_n^{(t)},
$$

where $A, B$ and $C$ are some positive constants. Now we get our final result:

$$
\inf_{t\leq T}\mathbb{E}\left[\left\|\nabla L_n\left(\boldsymbol{\omega}_n^{(t)}\right)\right\|^2\right] \leq \frac{1}{\sum_{t=0}^{T}\eta_t}\sum_{t=0}^{T}\eta_t\left\|\nabla L_n\left(\boldsymbol{\omega}_n^{(t)}\right)\right\|^2
$$

$$
\leq \mathcal{O}\left(\frac{1}{\sum_{t=0}^{T}\eta_t}\right) + \mathcal{O}\left(\frac{\sum_{t=0}^{T}\rho_n^{(t)}\eta_t}{\sum_{t=0}^{T}\eta_t}\right) + \mathcal{O}\left(\frac{\sum_{t=0}^{T}\xi_n^{(t)}\eta_t}{\sum_{t=0}^{T}\eta_t}\right) + \mathcal{O}\left(\frac{\sum_{t=0}^{T}\eta_t^2}{\sum_{t=0}^{T}\eta_t}\right).
$$

It is simple to verify that $\frac{1}{\sum_{t=0}^{T} \eta_t} \to 0$ and $\frac{\sum_{t=0}^{T} \eta_t^2}{\sum_{t=0}^{T} \eta_t} \to 0$ if $T \to \infty$. As for $\frac{\sum_{t=0}^{T} \rho_n^{(t)} \eta_t}{\sum_{t=0}^{T} \eta_t}$, according to the Cauchy-Schwartz inequality, we have

$$
\begin{aligned}
\sum_{t=0}^{T} \rho_n^{(t)} \eta_t &= \sum_{t=0}^{T} \sqrt{\rho_n^{(t)}} \left( \sqrt{\rho_n^{(t)}} \eta_t \right) \\
&\leq \sqrt{\left( \sum_{t=0}^{T} \rho_n^{(t)} \right) \left( \sum_{t=0}^{T} \rho_n^{(t)} \eta_t^2 \right)} \\
&\leq \sqrt{\left( \sum_{t=0}^{T} \rho_n^{(t)} \right) \left( \sum_{t=0}^{T} \rho_n^{(t)} \right) \left( \sum_{t=0}^{T} \eta_t^2 \right)} \\
&< \infty.
\end{aligned}
$$

Hence, we also have $\frac{\sum_{t=0}^{T} \rho_n^{(t)} \eta_t}{\sum_{t=0}^{T} \eta_t} \to 0$ if $T \to \infty$. Similarly, we get the same result for $\frac{\sum_{t=0}^{T} \xi_n^{(t)} \eta_t}{\sum_{t=0}^{T} \eta_t}$. In conclusion, we get the result in Section 3.1:

$$
\lim_{T \to \infty} \inf_{t \leq T} \mathbb{E} \left\| \nabla L_n \left( \boldsymbol{\omega}_n^{(t)} \right) \right\|^2 = 0. \tag{38}
$$

$\square$

# B    PROOFS OF THEOREM 2, 3 AND ADDITIONAL ANALYSIS

## B.1    PROOF OF THEOREM 2

**Theorem 2.** *Assume that $\tilde{l}_m(\boldsymbol{z}_m, y)$ is $\mu_m$-strongly convex in $\boldsymbol{z}_m$ for each module $m$. If each module $m \leq M$ has local $(\epsilon_{m-1}, c_m)$-robustness in $l_m(\boldsymbol{z}_{m-1}, y)$, and*

$$
\forall m \leq M, \quad \epsilon_m \geq \frac{g_m}{\mu_m} + \sqrt{\frac{2c_m}{\mu_m} + \frac{g_m^2}{\mu_m^2}}, \quad \text{where } g_m = \|\nabla_{\boldsymbol{z}_m} \tilde{l}_m(\boldsymbol{z}_m, y)\|, \tag{39}
$$

*then we can guarantee that the entire model has a joint $(\epsilon_0, c_M)$-robustness in $l(\boldsymbol{x}, y)$.*

*Proof.* We only need to prove the joint robustness of the concatenation of module $m$ and $(m + 1)$ given the local robustness of them separately, and then we can use deduction to get the joint robustness of the entire model given the local robustness of all modules.

For a module $m$ and any perturbation $\boldsymbol{\delta}_{m-1} \in \{\boldsymbol{\delta}_{m-1} : \|\boldsymbol{\delta}_{m-1}\| \leq \epsilon_{m-1}\}$ at its input, let $\boldsymbol{r} = f_m(\boldsymbol{z}_{m-1} + \boldsymbol{\delta}_{m-1}) - f_m(\boldsymbol{z}_{m-1})$. Given $\mu_m$-strongly convexity and $(\epsilon_{m-1}, c_m)$-robustness in $\tilde{l}_m(\boldsymbol{z}_m, y)$, we have

$$
\nabla_{\boldsymbol{z}_m} \tilde{l}_m(\boldsymbol{z}_m, y)^T \boldsymbol{r} + \frac{\mu_m}{2} \|\boldsymbol{r}\|^2 \leq \tilde{l}_m(\boldsymbol{z}_m + \boldsymbol{r}, y) - \tilde{l}_m(\boldsymbol{z}_m, y) \leq c_m \tag{40}
$$

$$
\Rightarrow \frac{\mu_m}{2} \left[ \left\| \boldsymbol{r} + \frac{\nabla_{\boldsymbol{z}_m} \tilde{l}_m(\boldsymbol{z}_m, y)}{\mu_m} \right\|^2 - \frac{\|\nabla_{\boldsymbol{z}_m} \tilde{l}_m(\boldsymbol{z}_m, y)\|^2}{\mu_m^2} \right] \leq c_m \tag{41}
$$

$$
\Rightarrow \left\| \boldsymbol{r} + \frac{\nabla_{\boldsymbol{z}_m} \tilde{l}_m(\boldsymbol{z}_m, y)}{\mu_m} \right\| \leq \sqrt{\frac{2c_m}{\mu_m} + \frac{\|\nabla_{\boldsymbol{z}_m} \tilde{l}_m(\boldsymbol{z}_m, y)\|^2}{\mu_m^2}} \tag{42}
$$

$$
\Rightarrow \|\boldsymbol{r}\| \leq \frac{\|\nabla_{\boldsymbol{z}_m} \tilde{l}_m(\boldsymbol{z}_m, y)\|}{\mu_m} + \sqrt{\frac{2c_m}{\mu_m} + \frac{\|\nabla_{\boldsymbol{z}_m} \tilde{l}_m(\boldsymbol{z}_m, y)\|^2}{\mu_m^2}} = \frac{g_m}{\mu_m} + \sqrt{\frac{2c_m}{\mu_m} + \frac{g_m^2}{\mu_m^2}}. \tag{43}
$$

And we know that

$$
\epsilon_m \geq \frac{g_m}{\mu_m} + \sqrt{\frac{2c_m}{\mu_m} + \frac{g_m^2}{\mu_m^2}} \geq \|\boldsymbol{r}\|, \tag{44}
$$

which gives us

$$\forall \boldsymbol{\delta}_{m-1} \in \{\boldsymbol{\delta}_{m-1} : \|\boldsymbol{\delta}_{m-1}\| \leq \epsilon_{m-1}\}, \|f_m(\boldsymbol{z}_{m-1} + \boldsymbol{\delta}_{m-1}) - f_m(\boldsymbol{z}_{m-1})\| \leq \epsilon_m. \quad (45)$$

With the $(\epsilon_m, c_{m+1})$-robustness of module $(m+1)$, we have the joint robustness of the concatenation of $m$ and $(m+1)$:

$$\forall \boldsymbol{\delta}_{m-1} \in \{\boldsymbol{\delta}_{m-1} : \|\boldsymbol{\delta}_{m-1}\| \leq \epsilon_{m-1}\},$$
$$l_{m+1}(f_m(\boldsymbol{z}_{m-1} + \boldsymbol{\delta}_{m-1}), y) - l_{m+1}(f_m(\boldsymbol{z}_{m-1}), y) \leq c_{m+1}. \quad (46)$$

$\square$

## B.2 PROOF OF THEOREM 3

**Theorem 3.** *Assume that $\tilde{l}_m(\boldsymbol{z}_m, y)$ and $\tilde{l}'_m(\boldsymbol{z}_m, y)$ are $\beta_m, \beta'_m$-smooth in $\boldsymbol{z}_m$ for a module $m$. If there exist $c_m$, $c'_m$, and $r \geq \sqrt{2 \frac{c_m + c'_m}{\beta_m + \beta'_m}}$, such that the auxiliary model has $(r, c_m)$-robustness in $\tilde{l}_m(\boldsymbol{z}_m, y)$, and the backbone network has $(r, c'_m)$-robustness in $\tilde{l}'_m(\boldsymbol{z}_m, y)$, then we have:*

$$\|\nabla_{\boldsymbol{w}_m} l - \nabla_{\boldsymbol{w}_m} l_m\| \leq \left\| \frac{\partial \boldsymbol{z}_m}{\partial \boldsymbol{w}_m} \right\| \sqrt{2(c_m + c'_m)(\beta_m + \beta'_m)}. \quad (47)$$

*Proof.* With the chain rule, we know that

$$\nabla_{\boldsymbol{w}_m} l - \nabla_{\boldsymbol{w}_m} l_m = \frac{\partial \boldsymbol{z}_m}{\partial \boldsymbol{w}_m} \frac{\partial (l - l_m)}{\partial \boldsymbol{z}_m} = \frac{\partial \boldsymbol{z}_m}{\partial \boldsymbol{w}_m} \frac{\partial (\tilde{l}'_m - \tilde{l}_m)}{\partial \boldsymbol{z}_m}, \quad (48)$$

and thus

$$\|\nabla_{\boldsymbol{w}_m} l - \nabla_{\boldsymbol{w}_m} l_m\| \leq \left\| \frac{\partial \boldsymbol{z}_m}{\partial \boldsymbol{w}_m} \right\| \left\| \frac{\partial (\tilde{l}'_m - \tilde{l}_m)}{\partial \boldsymbol{z}_m} \right\|. \quad (49)$$

We now need to find the upper bound of the second factor. We define $h(\boldsymbol{z}_m) = \tilde{l}'_m(\boldsymbol{z}_m, y) - \tilde{l}_m(\boldsymbol{z}_m, y)$, which is $(\beta_m + \beta'_m)$-smooth in $\boldsymbol{z}_m$. For any $\boldsymbol{\delta}_m, \|\boldsymbol{\delta}_m\| \leq r$, with the $(r, c_m)$-robustness in $\tilde{l}_m(\boldsymbol{z}_m, y)$ and $(r, c'_m)$-robustness in $\tilde{l}'_m(\boldsymbol{z}_m, y)$, we have

$$|h(\boldsymbol{z}_m + \boldsymbol{\delta}_m) - h(\boldsymbol{z}_m)| \leq |\tilde{l}_m(\boldsymbol{z}_m + \boldsymbol{\delta}_m, y) - \tilde{l}_m(\boldsymbol{z}_m, y)| + |\tilde{l}'_m(\boldsymbol{z}_m + \boldsymbol{\delta}_m, y) - \tilde{l}'_m(\boldsymbol{z}_m, y)|$$
$$\leq c_m + c'_m. \quad (50)$$

And with the $(\beta_m + \beta'_m)$-smoothness, we know that

$$\left( \frac{\partial h(\boldsymbol{z}_m)}{\partial \boldsymbol{z}_m} \right)^T \boldsymbol{\delta}_m - \frac{\beta_m + \beta'_m}{2} \|\boldsymbol{\delta}_m\|^2 \leq h(\boldsymbol{z}_m + \boldsymbol{\delta}_m) - h(\boldsymbol{z}_m) \leq c_m + c'_m. \quad (51)$$

The maximum of the LHS is achieved when $\boldsymbol{\delta}_m^* = \frac{1}{\beta_m + \beta'_m} \frac{\partial h(\boldsymbol{z}_m)}{\partial \boldsymbol{z}_m}$, and thus we get

$$\frac{1}{2(\beta_m + \beta'_m)} \left\| \frac{\partial h(\boldsymbol{z}_m)}{\partial \boldsymbol{z}_m} \right\|^2 \leq c_m + c'_m \quad (52)$$

$$\Rightarrow \left\| \frac{\partial h(\boldsymbol{z}_m)}{\partial \boldsymbol{z}_m} \right\| \leq \sqrt{2(c_m + c'_m)(\beta_m + \beta'_m)}. \quad (53)$$

To check the achievability of this maximum, we have

$$\|\boldsymbol{\delta}_m^*\| = \frac{1}{\beta_m + \beta'_m} \left\| \frac{\partial h(\boldsymbol{z}_m)}{\partial \boldsymbol{z}_m} \right\| \leq \sqrt{2 \frac{c_m + c'_m}{\beta_m + \beta'_m}} \leq r. \quad (54)$$

Thus, we get our final result

$$\|\nabla_{\boldsymbol{w}_m} l - \nabla_{\boldsymbol{w}_m} l_m\| \leq \left\| \frac{\partial \boldsymbol{z}_m}{\partial \boldsymbol{w}_m} \right\| \sqrt{2(c_m + c'_m)(\beta_m + \beta'_m)}. \quad (55)$$

$\square$

### B.3 CASE STUDY: LINEAR AUXILIARY OUTPUT MODEL

For a linear auxiliary output model $\boldsymbol{\theta}_m = \{\boldsymbol{W}_m, \boldsymbol{b}_m\}$, the cross-entropy loss is given as

$$\tilde{l}(\boldsymbol{z}_m, \boldsymbol{y}) = \mathcal{L}(\sigma(\boldsymbol{W}_m^T \boldsymbol{z}_m + \boldsymbol{b}_m), \boldsymbol{y}), \tag{56}$$

where $\mathcal{L}(\boldsymbol{p}, \boldsymbol{y}) = -\sum_{i=1} y_i \log(p_i)$ and $\sigma(\boldsymbol{q})_i = \exp(q_i)/(\sum_j \exp(q_j))$ are cross-entropy loss and softmax function respectively. Let $\boldsymbol{p}_m = \sigma(\boldsymbol{W}_m^T \boldsymbol{z}_m + \boldsymbol{b}_m)$, we know that

$$\nabla_{\boldsymbol{z}_m} \tilde{l}(\boldsymbol{z}_m, \boldsymbol{y}) = \boldsymbol{W}_m(\boldsymbol{p}_m - \boldsymbol{y}), \tag{57}$$

and

$$\boldsymbol{H}_m = \nabla_{\boldsymbol{z}_m}^2 \tilde{l}(\boldsymbol{z}_m, \boldsymbol{y}) = \boldsymbol{W}_m \boldsymbol{J}_m \boldsymbol{W}_m^T, \tag{58}$$

where

$$\boldsymbol{J}_m = \mathrm{diag}(\boldsymbol{p}_m) - \boldsymbol{p}_m \boldsymbol{p}_m^T \tag{59}$$

is the Jacobi of the softmax function. We have the following properties related to the robustness and objective consistency in Theorem 2 and Theorem 3:

1. (First Order Property) Smaller $\|\boldsymbol{W}_m\|$ leads to smaller $g_m$ and $c_m$.

$$g_m = \|\nabla_{\boldsymbol{z}_m} \tilde{l}_m(\boldsymbol{z}_m, y)\| = \|\boldsymbol{W}_m(\boldsymbol{p}_m - \boldsymbol{y})\| \leq \sqrt{2}\|\boldsymbol{W}_m\|, \tag{60}$$

$$c_m = \max_{\|\boldsymbol{\delta}_m\| \leq r} |\tilde{l}_m(\boldsymbol{z}_m + \boldsymbol{\delta}_m, y) - \tilde{l}_m(\boldsymbol{z}_m, y)| \leq \sqrt{2}r\|\boldsymbol{W}_m\|. \tag{61}$$

2. (Second Order Property) Smaller $\|\boldsymbol{W}_m\|_F$ leads to smaller $\mu_m$ and $\beta_m$.

$$\sum_i \lambda_i(\boldsymbol{H}_m) = \mathrm{tr}(\boldsymbol{H}_m) = \mathrm{tr}(\boldsymbol{W}_m \boldsymbol{J}_m \boldsymbol{W}_m^T) = \mathrm{tr}(\boldsymbol{W}_m^T \boldsymbol{W}_m \boldsymbol{J}_m)$$

$$\leq \|\boldsymbol{W}_m\|_F^2 (\sum_j p_{m,j} - p_{m,j}^2), \tag{62}$$

where $\lambda_i(\boldsymbol{H}_m)$ means the eigenvalues of $\boldsymbol{H}_m$ in increasing order. $\lambda_1(\boldsymbol{H}_m) = \mu_m$ and $\lambda_{-1}(\boldsymbol{H}_m) = \beta_m$.

We notice that when increasing $\lambda_m$, namely, decreasing $\|\boldsymbol{W}_m\|$ and $\|\boldsymbol{W}_m\|_F$, we will decrease $g_m, c_m, \mu_m$ and $\beta_m$. According to Theorem 2, smaller $g_m$ will lead to stronger robustness while smaller $\mu_m$ will lead to weaker robustness. And according to Theorem 3, smaller $c_m$ and $\beta_m$ can lead to smaller objective inconsistency and thus better natural accuracy.

## C EXPERIMENT SETTINGS AND DETAILS

We run all the experiments on a sever with a single NVIDIA TITAN RTX GPU and an Intel Xeon Gold 6254 CPU.

### C.1 DETAILS OF BASELINES

**FedDynAT (Shah et al., 2021).** FedDynAT proposes to use an annealing number of local training iterations to alleviate the slow convergence issue of Federated Adversarial Training (FAT) (Zizzo et al., 2020). More specifically, they anneal the number of local training iterations as $\tau_t = \tau_0 \gamma_E^{t/F_E}$ where $\tau_t$ is the number of local training iterations at round $t$, $\gamma_E$ is the decay rate and $F_E$ is the decay period. When implementing FedDynAT, we use FedNOVA instead of FedCurv (Shoham et al., 2019) to avoid extra communication in our resource-constrained settings.

**FedRBN (Hong et al., 2021).** FedRBN adopts Dual Batch Normalization (DBN) layers (Xie et al., 2020) with two sets of batch normalization (BN) statistics for clean samples and adversarial samples respectively. When propagating the robustness from the clients who perform AT to the clients

Table 4: The hyperparameters of FADE. The last module does not have an auxiliary model since it directly uses the loss of the backbone network, thus it does not have a weight decay hyperparameter.

| Model | Optimizer | Module 1 | | | Module 2 | | | Module 3 | | |
|---|---|---|---|---|---|---|---|---|---|---|
| | | $\epsilon_0$ | $\alpha_0$ | $\lambda_1$ | $\epsilon_1$ | $\alpha_1$ | $\lambda_2$ | $\epsilon_2$ | $\alpha_2$ | $\lambda_3$ |
| 2-module CNN-7 | FedNOVA | 0.15 | 0.03 | 0.003 | 0.06 | 0.012 | n/a | n/a | n/a | n/a |
| FMNIST | FedBN | 0.15 | 0.03 | 0.03 | 0.12 | 0.024 | n/a | n/a | n/a | n/a |
| 2-module VGG-11 | FedNOVA | $8/255$ | $2/255$ | 0.1 | $3/255$ | $0.75/255$ | n/a | n/a | n/a | n/a |
| CIFAR-10 | FedBN | $8/255$ | $2/255$ | 0.003 | $3/255$ | $0.75/255$ | n/a | n/a | n/a | n/a |
| 3-module VGG-11 | FedNOVA | $8/255$ | $2/255$ | 0.003 | $4/255$ | $1/255$ | 0.003 | $3/255$ | $0.75/255$ | n/a |
| CIFAR-10 | FedBN | $8/255$ | $2/255$ | 0.001 | $4/255$ | $1/255$ | 0.001 | $3/255$ | $0.75/255$ | n/a |
| Mixing VGG-11 | FedNOVA | $8/255$ | $2/255$ | 0.01 | $4/255$ | $1/255$ | 0.01 | $3/255$ | $0.75/255$ | n/a |
| CIFAR-10 | FedBN | $8/255$ | $2/255$ | 0.001 | $4/255$ | $1/255$ | 0.001 | $3/255$ | $0.75/255$ | n/a |

who perform ST, they use the adversarial BN statistics of AT clients to evaluate the adversarial BN statistics of ST clients as follows:

$$\mu_{\text{ST}}^a = \mu_{AT}^a + \lambda_{\text{RBN}}(\mu_{\text{ST}}^n - \mu_{AT}^n), \tag{63}$$

$$(\sigma_{\text{ST}}^a)^2 = (\sigma_{AT}^a)^2 \left[ \frac{(\sigma_{\text{ST}}^n)^2}{(\sigma_{AT}^n)^2 + \epsilon} \right]^{\lambda_{\text{RBN}}} \tag{64}$$

where $\mu_{\text{ST}}^a$ and $\mu_{\text{ST}}^n$ are the means in adversarial BN and natural BN respectively on a ST client, and $(\sigma_{\text{ST}}^a)^2$ and $(\sigma_{\text{ST}}^n)^2$ are the variances. Similarly, for AT clients we have $\mu_{AT}^a$, $\mu_{AT}^n$, $(\sigma_{AT}^a)^2$ and $(\sigma_{AT}^n)^2$. $\lambda_{\text{RBN}}$ is the hyperparameter and $\epsilon$ is a small constant. With these evaluations of the adversarial BN statistics, the ST clients can also attain some adversarial robustness without performing AT.

## C.2 HYPERPARAMETERS

**Hyperparameters of FL**  We partition the whole FMNIST and CIFAR-10 dataset (including the training set and the test set) onto $N = 100$ clients, with a training-validation ratio the same as the ratio of the training set to the test set. For global FL (FedNOVA), the validation sets on all clients are I.I.D.. For personalized FL (FedBN), we make the validation set on each client have the same distribution as the training set on that client (i.e., Non-I.I.D.). We report the averaged validation accuracy over all clients in our experiments.

We set the number of initial local training iterations as $\tau_0 = 40$ for both FMNIST and CIFAR-10, and the local batch size is set to be $B = 50$. We use the same trick as Shah et al. (2021) that we gradually decrease the number of local training iterations. When training with FedNOVA, the maximal number of communication rounds $T = 500$ for FMNIST and $T = 2000$ for CIFAR-10. When training with FedBN, the maximal number of communication rounds $T = 150$ for FMNIST and $T = 500$ for CIFAR-10. We use the SGD optimizer with a constant learning rate $\eta = 0.01$ and momentum 0.9 for all the experiments.

**Hyperparameters of AT**  We adopt $\epsilon_0 = 0.15$ and $\alpha_0 = 0.03$ for FMNIST, and we use $\epsilon_0 = 8/255$ and $\alpha_0 = 2/255$ for CIFAR-10. We use PGD with $T = 10$ iterations for training and testing in all our experiments. Following Zizzo et al. (2020), we use a warmup phase with only standard training before performing any AT in all the experiments. When training with FedNOVA, the length of the warmup phase is set to be 50 for FMNIST and 400 for CIFAR-10. When training with FedBN, the length of the warmup phase is set to be 15 for FMNIST and 200 for CIFAR-10.

**Hyperparameters of FADE**  Table 4 summarizes the $\lambda_m$ and $\epsilon_{m-1}$ that we used in the experiments in Section 4.3. When tuning both $\epsilon_m$ and $\lambda_m$, we adopt the overall accuracy on both clean and adversarial samples as the criterion, which can be written as $A = 0.4A_n + 0.6A_a$ where $A_n$ and $A_a$ are natural accuracy and adversarial accuracy respectively. When determining the feature perturbation $\epsilon_1$ and $\epsilon_2$, we perform linear search for the optimal discount factors $d_1, d_2 \in [0.0, 1.0]$ such that $\epsilon_1 = d_1\epsilon_0$ and $\epsilon_2 = d_2\epsilon_0$. When determining the weight decay hyperparameter $\lambda_m$, we select the optimal $\lambda_m \in \{0.0001, 0.0003, 0.001, 0.003, 0.01, 0.03, 0.1\}$.

**Hyperparameters of FedDynAT**  We set the decay rate $\gamma_E = 0.9$, and the decay period $F_E$ is always set to be $1/10$ of the maximal number of communication rounds $T$.

Table 5: The model architecture and the 2-module partition of CNN-7.

| Module | Layer | Details | # Parameters |
|--------|-------|---------|--------------|
| 1 | 1 | Conv2D (8, kernel size = 3, padding = 1, stride = 1) BN2D, ReLU, MaxPool2D (kernel size = 2, stride = 2) | 15.312 k |
| | 2 | Conv2D (16, kernel size = 3, padding = 1, stride = 1) BN2D, ReLU, MaxPool2D (kernel size = 2, stride = 2) | |
| | 3 | Conv2D (32, kernel size = 3, padding = 1, stride = 1) BN2D, ReLU | |
| | 4 | Conv2D (32, kernel size = 3, padding = 1, stride = 1) BN2D, ReLU, MaxPool2D (kernel size = 2, stride = 2) | |
| | A1 | FC (288, 10) | 2.890 k |
| 2 | 5 | Conv2D (64, kernel size = 3, padding = 0, stride = 1) BN2D, ReLU | 23.562 k |
| | 6 | FC (64, 64), BN1D, ReLU | |
| | 7 | FC (64, 10) | |

Table 6: The model architecture and the 2-module partition of VGG-11.

| Module | Layer | Details | # Parameters |
|--------|-------|---------|--------------|
| 1 | 1 | Conv2D (64, kernel size = 3, padding = 1, stride = 1) BN2D, ReLU, MaxPool2D (kernel size = 2, stride = 2) | 4.504 M |
| | 2 | Conv2D (128, kernel size = 3, padding = 1, stride = 1) BN2D, ReLU, MaxPool2D (kernel size = 2, stride = 2) | |
| | 3 | Conv2D (256, kernel size = 3, padding = 1, stride = 1) BN2D, ReLU | |
| | 4 | Conv2D (256, kernel size = 3, padding = 1, stride = 1) BN2D, ReLU, MaxPool2D (kernel size = 2, stride = 2) | |
| | 5 | Conv2D (512, kernel size = 3, padding = 1, stride = 1) BN2D, ReLU | |
| | 6 | Conv2D (512, kernel size = 3, padding = 1, stride = 1) BN2D, ReLU, MaxPool2D (kernel size = 2, stride = 2) | |
| | A1 | MaxPool2D (kernel size = 2, stride = 2), FC (512, 10) | 0.005 M |
| 2 | 7 | Conv2D (512, kernel size = 3, padding = 1, stride = 1) BN2D, ReLU | 5.254 M |
| | 8 | Conv2D (512, kernel size = 3, padding = 1, stride = 1) BN2D, ReLU, MaxPool2D (kernel size = 2, stride = 2) | |
| | 9 | FC (512, 512), BN1D, ReLU | |
| | 10 | FC (512, 512), BN1D, ReLU | |
| | 11 | FC (512, 10) | |

**Hyperparameters of FedRBN** Following Hong et al. (2021), we adopt the same setting where $\lambda_{\text{RBN}} = 0.1$. We loose the requirement of a noise detector and allow an optimal noise detector for FedRBN such that it can always use the correct BN statistics during test (This makes its robustness stronger than that with a real noise detector).

C.3 MODEL ARCHITECTURES AND MODEL PARTITIONS

The model architectures and model partitions we use in our experiments are shown in Table 5, 6 and 7. We skip all the Batch Normalization layers in the model when training with FedNOVA. We also show the number of parameters in the tables. We could see that in most case the auxiliary models are small enough and they only introduce negligible extra parameters and computation. Therefore it will not increase the memory and computing power requirements on resource-constraint devices.

Table 7: The model architecture and the 3-module partition of VGG-11.

| Module | Layer | Details | # Parameters |
|---|---|---|---|
| 1 | 1 | Conv2D (64, kernel size = 3, padding = 1, stride = 1) 
 BN2D, ReLU, MaxPool2D (kernel size = 2, stride = 2) | 0.962 M |
| | 2 | Conv2D (128, kernel size = 3, padding = 1, stride = 1) 
 BN2D, ReLU, MaxPool2D (kernel size = 2, stride = 2) | |
| | 3 | Conv2D (256, kernel size = 3, padding = 1, stride = 1) 
 BN2D, ReLU | |
| | 4 | Conv2D (256, kernel size = 3, padding = 1, stride = 1) 
 BN2D, ReLU, MaxPool2D (kernel size = 2, stride = 2) | |
| | A1 | MaxPool2D (kernel size = 2, stride = 2), FC (1024, 10) | 0.010 M |
| 2 | 5 | Conv2D (512, kernel size = 3, padding = 1, stride = 1) 
 BN2D, ReLU | 3.542 M |
| | 6 | Conv2D (512, kernel size = 3, padding = 1, stride = 1) 
 BN2D, ReLU, MaxPool2D (kernel size = 2, stride = 2) | |
| | A2 | MaxPool2D (kernel size = 2, stride = 2), FC (512, 10) | 0.005 M |
| 3 | 7 | Conv2D (512, kernel size = 3, padding = 1, stride = 1) 
 BN2D, ReLU | 5.254 M |
| | 8 | Conv2D (512, kernel size = 3, padding = 1, stride = 1) 
 BN2D, ReLU, MaxPool2D (kernel size = 2, stride = 2) | |
| | 9 | FC (512, 512), BN1D, ReLU | |
| | 10 | FC (512, 512), BN1D, ReLU | |
| | 11 | FC (512, 10) | |

