# OpenReview forum: "FADE: Enabling Large-Scale Federated Adversarial Training on Resource-Constrained Edge Devices"
_ICLR.cc/2023/Conference — Submitted to ICLR 2023_

### Official Review · Reviewer_8e8c · 2022-10-22

**Confidence:** 4
**Correctness:** 3
**Technical Novelty And Significance:** 3
**Empirical Novelty And Significance:** 3
**Recommendation:** 6

**Clarity, Quality, Novelty And Reproducibility:**

Quality: The quality of this paper is good. The authors propose to exploit Decoupled Learning in Federated Adversarial Learning and provide the corresponding trade-off technique and theoretical guarantee.
Clarity: The clarity of this paper needs to be improved. In section 3.1, the meaning of Eq. 9 conflicts with the above description of it and the meaning of Figure 2. The authors said that the proposed method can ensure flexible model partitions, that is to say, different clients have different model partitions, as shown in Figure 2. However, what Eq. 9 means is only for the case that different clients have the same model partition. Additionally, there are some typos and unclear writings (See in “Strength And Weaknesses”).
Originality: The originality of this paper is limited. The authors incorporate Decoupled Learning into Federated Adversarial Learning and propose a weight decay strategy. However, analyses for the case that different clients have different module partitions are lacking.


**Strength And Weaknesses:**

This paper focuses on federated adversarial training (FAT) on resource-constrained edge devices, which is an important and less researched topic. The authors incorporate Decoupled Learning (DL) into FAT naturally, analyze and solve the resulting problems theoretically and technologically. Specially, they propose FADE, offer a theoretical guarantee for convergence and adversarial robustness and provide a technique to alleviate objective inconsistency and achieve better accuracy-robustness balance. However, I still have some concerns and suggestions:
1.	In section 3.1, the meaning of Eq. 9 conflicts with the above description of it and the meaning of Figure 2. The authors said that the proposed method can ensure flexible model partitions, that is to say, different clients have different model partitions, as shown in Figure 2. However, what Eq. 9 means is only for the case that different clients have the same model partition. In fact, from my perspective, throughout the paper, the analyses and experiments are constructed on the assumption that different clients have the same model partition. Only the experiment “FADE (Mixing)” assumes that different clients have different model partitions, yet the detail of implementation is unclear. For the case that different clients have different model partitions, I wonder how to aggregate model parameters when the data across clients are heterogenous.
2.	When talking about Decoupled Greedy Learning (DGL), the authors can give some explanations and examples of locally supervised loss. As a reader, why and how to use locally supervised loss is unclear.
3.	In related work, the authors should give some introduction of FedDynAT and FedRBN, both of which are compared in the experiments.
4.	There are some typos. In the third paragraph of section 2, “Decouled Greedy Learning (DGL)” should be “Decoupled Greedy Learning (DGL)”. In line 8 and 9 of Algorithm 1, $\Theta^{(t+1)}_{m_k^t,k}$ should be $\Theta^{(t+1)}_{m_k^t}$ to be consistent with the notations in section “Decoupled Greedy Learning (DGL)”.


**Summary Of The Paper:**

This paper studies how to perform federated adversarial training (FAT) on resource-constrained edge devices. The authors propose Federated Adversarial Decoupled Learning (FADE) to decouple the entire model into small modules for feasible FAT. They also offer a theoretical guarantee for convergence and adversarial robustness, based on which, a technique to alleviate objective inconsistency and achieve better accuracy-robustness balance is proposed.

**Summary Of The Review:**

This paper focuses on federated adversarial training (FAT) on resource-constrained edge devices. The authors propose to exploit Decoupled Learning in Federated Adversarial Learning and provide the corresponding trade-off technique and theoretical guarantee. However, analyses for the case that different clients have different module partitions are lacking and the clarity needs to be improved (See in “Strength And Weaknesses”).

==I read the authors' rebuttal and comments from other PCs, I updated my scorings for its merits and shortcominings.

---

> ### Author Response · Authors · 2022-11-10
> **Response to The Review**
>
> Thank you for your valuable review. We have made some revisions (in blue color) in the new version of our paper according to your comments. And we try to summarize your questions and answer them below.
>
> Q1. Eq. 9 conflicts with the argument that we can adopt different model partitions on different clients.
>
> A1. To make our statements simpler and clearer, we rewrite the notations in Section 3.1 to become the same as those we used in the proof of Theorem 1 in Appendix A. More specifically, we discuss the update and aggregation of a layer $n$ instead of a single parameter $\boldsymbol{\omega}$. These two notations are equivalent because one single layer is the "atom" in decoupled learning and cannot be further decoupled.
>
> In previous versions, the confusing point may be the sentence "each client $k$ randomly samples a module $m_k^t$ from their own model partitions". There is a hidden condition that we did not state explicitly in this sentence, that is $m_i^t$ and $m_j^t$ can be sampled from different model partitions if client $i$ and client $j$ have different model partitions. More rigorously, let us denote the set of all modules on client $k$ as $\mathbb{M}_k$, and $m_k^t$ is sampled from $\mathbb{M}_k$ in each round. For two different client $i$ and $j$, $\mathbb{M}_i\neq\mathbb{M}_j$ if client $i$ is using a different model partition from client $j$. We take the case in Figure 2 as an example. The first client with 4 modules has $\mathbb{M}_1 =$ {{$n_1$,$n_2$,$A_1$},{$n_3$,$n_4$,$A_2$},{$n_5$,$n_6$,$A_3$},{$n_7$,$n_8$,$A_4$}} where $n_i$ means the $i$-th layer in backbone network and $A_i$ means the auxiliary output model of the $i$-th module. At the same time, the second client with 2 modules has $\mathbb{M}_2$ = {{$n_1$,$n_2$,$n_3$,$n_4$,$A_2$},{$n_5$,$n_6$,$n_7$,$n_8$,$A_4$}}, which is different from $\mathbb{M}_1$.
>
> According to the discussion above, Eq. 9 can actually represent the aggregation rule when different clients are using different model partitions. We have explicitly pointed it out in Section 3.1 of our new version.
>
> Q2. Experiments do not adopt different model partitions on different clients.
>
> A2. In all of "FADE (2 Modules)", "FADE (3 Modules)" and "FADE (Mixing)", we are adopting different model partitions on different clients. Those resource-sufficient clients that can afford joint AT have only one module (i.e., the entire model), which is also one kind of model partition as shown in Figure 2. The other resource-insufficient clients use the 2-module partition or the 3-module partition, which are different from the model partition used by the resource-sufficient clients. We conduct our experiments in Figure 4 with different proportions of resource-sufficient clients. In Tables 2 and 3, we fix the proportion of the resource-sufficient clients to be 20\%, while 80\% clients are using the 2/3-module partition in "FADE (2/3 Modules)". In "FADE (Mixing)", the clients with one module, two modules and three modules are mixed in a ratio of 2:3:5. Furthermore, we also conducted experiments in Figure 4 to show the robustness when training with different proportions of the resource-sufficient clients. We provide these details in Section 4.3.
>
> Q3. Why and how to use locally supervised loss?
>
> A3. The locally supervised loss is used so that we do not need to go through the whole model to get the joint loss $l$ and perform backpropagation. As we illustrate in Figure 1, we can immediately get the locally supervised loss $l_m$ of module $m$ after going through this module. When performing training on this module, we will use this locally supervised loss $l_m$ to train instead of the joint loss $l$ output by the entire model. In summary, the locally supervised loss is the key to decoupling the model and training each module separately. We have highlighted this point in Section 2 of our new version.
>
> Q4. More descriptions of the baselines are needed.
>
> A4. In FedDynAT, the author proposes to anneal the number of local training iterations to attain better performance. In FedRBN, the author adopts dual batch normalization and propagates the robustness in the batch normalization statistics from clients who can afford AT to those who cannot. We have included more descriptions of them in Appendix C.1 in our new version.
>
> Q5. $\Theta_{m_k^t,k}$ should be $\Theta_{m_k^t}$ in Algorithm 1.
>
> A5. We use $\Theta_{m_k^t,k}$ with subscript $k$ to denote the local model trained by client $k$ before aggregation in line 8 and line 9 of Algorithm 1. This notation is inherited from that of FL in Section 2. Notice that in the description of DGL in Section 2, we are following the original paper of DGL and discussing it in a centralized training setting, where we do not need to use $\Theta_{m_k^t,k}$ to distinguish the local models trained on different clients.

---

### Official Review · Reviewer_iHWT · 2022-10-24

**Confidence:** 4
**Correctness:** 3
**Technical Novelty And Significance:** 3
**Empirical Novelty And Significance:** 3
**Recommendation:** 6

**Clarity, Quality, Novelty And Reproducibility:**

The paper presents a novel idea of combing decoupled learning with adversarial training. The details of the proposed framework need further clarification.

**Strength And Weaknesses:**

Strengths:
1. The paper aims to address the challenges of adversarial training in resource-constrained federated learning, which is an important and timely problem.
2. FADE offers theoretical evidence for the convergence and adversarial robustness of the proposed framework.
3. The experimental results show that FADE reduces the computational cost while maintaining model robustness.

Weaknesses:
1. The proposed federated decoupled learning is unclear. It seems the paper only addresses the update locking using the auxiliary output model.  Forward passing is still required in federated decoupled learning. For example, in the second row of Figure 2, the model training still needs the outputs from the previous layers.
2. The min-max optimization problem (Eq. 11) requires to update $\delta_{m-1}$ to maximize the loss function. However, without accessing the first few layers, how do we update $\delta_{m-1}$?
3. The paper investigates two recent federated learning algorithms FedNOVA and FedBN. It would be nice to consider FedAVG as another baseline.
4. The paper considers 2-module and 3-module FADE. Can we split the model into more modules for extremely low-end devices?

Minor: Page 3. Decouled Greedy Learning -> Decouple Greedy Learning


**Summary Of The Paper:**

This paper proposes an adversarial training framework for resource-constrained federated learning. The proposed framework decouples the entire model into small modules to fit into the edge device memory. Adversarial training is only performed on a single module in each communication round.

**Summary Of The Review:**

The proposed framework is quite novel. The experimental results look promising. The paper presents theoretical proof for convergence and adversarial robustness. However, the details of the proposed framework need further clarification.

Thank you for the authors' responses. While the responses have addressed most of my concerns, I still have a concern about the use of forward passing in the first training round, which may impact the efficiency of the proposed method. I also agree with other reviewers that the paper's novelty may be somewhat limited as it simply applies decoupled learning in federated adversarial learning.

---

> ### Author Response · Authors · 2022-11-10
> **Response to The Review**
>
> Thank you for your valuable review. We have made some revisions (in blue color) in the new version of our paper according to your comments. And we try to summarize your questions and answer them below.
>
> Q1. Federated Decoupled Learning requires the forward passing through previous layers.
>
> A1. We only need to perform one forward passing through previous layers at the beginning of each training round. Then the outputs $z_{m-1}$ of the previous layers will be fixed during the training of module $m$ in this round, and the previous layers can be released. The one-time forward passing through previous layers does not increase the memory requirement and only consumes few computations compared to dozens of training iterations in each training round in FL. And for those devices who cannot load the entire model for forward passing because of small memory, at each time they can load one module and perform forward passing through it, and then they release this module before loading the next module and performing forward passing again. This will not introduce extra communication or computation consumption compared to loading the entire model at once.
>
> We believe that the confusing point may come from the wrong notations in the description of DGL in Section 2, where we mistakenly used $t$ to denote an iteration instead of an epoch. We have corrected it in our new version and we use $t$ to denote an epoch (in DGL) or a training round (in FDL) throughout our paper.
>
> Q2. Is the update of $\delta_{m-1}$ related to previous layers?
>
> A2. As we mentioned above, we will fix the input $z_{m-1}$ during the training of module $m$. We notice that $\delta_{m-1}$ is added on this fixed input $z_{m-1}$, thus it is independent with the previous layers. The update of $\delta_{m-1}$ is using PGD with the locally supervised loss $l_m$ of module $m$, which is only related to the parameters of module $m$.
>
> Q3. Why don't we consider FedAvg in our experiments?
>
> A3. Since FedNOVA is one of the state-of-the-art FL optimizers, it is shown to get better results than FedAvg. Besides, when the numbers of training iterations are the same on all clients, FedNOVA is the same as FedAvg.
>
> Q4. Can we split the model into more modules?
>
> A4. In theory, we can make each module as small as one layer, as we do not make any assumptions about the number of modules in our theory about convergence (Theorem 1) or robustness (Theorem 2). However, because of the characteristics of the VGG-like models (such as the CNN-7 and VGG-11 we use) that most parameters are concentrated in a small number of layers, we can only reduce the memory requirement marginally by further increasing the number of modules. For example, the 5th layer of CNN-7 contains 47\% parameters of the entire model, which means that the smallest memory requirement we can attain by increasing the number of modules in CNN-7 is 53\%. In this paper, we are focusing on developing a general algorithm that could be run on all models. Designing a model with a more uniform parameter distribution is beyond the topic of this paper, and we would like to leave it as a future work.

---

### Official Review · Reviewer_xYLU · 2022-10-24

**Confidence:** 4
**Correctness:** 3
**Technical Novelty And Significance:** 3
**Empirical Novelty And Significance:** 2
**Recommendation:** 6

**Clarity, Quality, Novelty And Reproducibility:**

Clarity: This paper is well organized, and most clarifications are clear and well-supported. However, the current improvement seems to be not significant, and more experiments on other challenging datasets or real-world datasets are also encouraged to be conducted, which can make the claims more convincing.

Quality: The presentation of the current version is of high quality with sufficient theoretical analysis. However, the relationship of the current theoretical analysis with algorithm 1 is not very clear.

Novelty: The novelty of the proposed decoupled learning has limited novelty compared with the original DGL. the method part could be further improved by highlighting its unique points for this problem that is different from related strategies.

Reproducibility: This paper has provided detailed information about the experimental settings. It could be better if the author could open-source its code later.


**Strength And Weaknesses:**

Strength:
1. The focused problem is very practical and significant to push federated adversarial training towards a more real-world scenario.
2. The proposed FADE shows empirical effectiveness in the experiments on FMNIST and CIFAR-10 datasets.
3. The corresponding theoretical analysis on the convergence of the proposed FADE is provided and seems to be correct.

Weaknesses:
1. The technical novelty of the proposed method is limited compared to DGL. Without a clear technical motivation to improve DGL, the proposed FADE seems to be heuristical. The presentation could further highlight the technical difficulty and how FADE address that.
2. The experiment part of the current version is weak. The performance gain of the proposed method is not significant and still suffers from severe accuracy-robustness tradeoff. Some quantitative experiments which can reflect the degree of resource-constrained edge devices seem to be missing, which can directly reflect the performance limitation of either previous or the proposed methods. Since the title is called "large-scale", some large-scale datasets can be involved to make the empirical results more convincing. It also lacks the appropriate ablation study to understand the proposed algorithmic component better. It needs further evidence or clear presentation to show the relationship of the effectiveness with the improved algorithm.
3. The current theoretical analysis indeed provides the convergence guarantee of the algorithm. Could the author provide more discussion about the relationship between the theoretical results with the resource-constrained setting?



**Summary Of The Paper:**

This paper focuses on federated adversarial training, especially learning in a resource-constrained setting. To be specific, the high demand for memory capacity and computational power makes federated adversarial training infeasible in the resource-constrained setting. To overcome this issue, this paper proposes Federated Adversarial Decoupled Learning (FADE), which allows a more flexible model partition on devices to fit the resource budgets. Correspondingly, the authors provide the theoretical guarantee for the convergence of the proposed algorithm and conduct experiments to empirically verify the effectiveness of FADE.

**Summary Of The Review:**

Overall, I think this paper proposed a promising method for federated adversarial training under resource-constrained edge devices, with corresponding theoretical analysis. However, the presentation part and the experimental parts can be further improved.

---

> ### Author Response · Authors · 2022-11-10
> **Response to The Review**
>
> Thank you for your valuable review. We have made some revisions (in blue color) in the new version of our paper according to your comments. And we try to summarize your questions and answer them below.
>
> Q1. Novelty is not significant compared to DGL.
>
> A1. We highlight two differences between FADE and DGL in Section 1:
>
> 1. FADE allows different model partitions on different clients, which makes it more general and more suitable for FL than DGL. In DGL, they only consider the same model partitions on all the devices, and the convergence analysis is also restricted in this case. This is not practical in FL when different devices have different resource budgets. Our Theorem 1 guarantees that FADE can converge with different model partitions on different clients.
>
> 2. FADE is the first algorithm to complement theoretically guaranteed joint robustness into DGL. Notice that it is not trivial to guarantee joint robustness when combining AT and DGL, because performing AT in locally supervised losses does not necessarily lead to robustness in the joint loss. Our Theorem 2 gives the sufficient condition to guarantee joint robustness in FADE.
>
> Q2. The improvement seems to be marginal.
>
> A2. Our improvement mainly lies in three aspects.
>
> 1. We largely reduce the memory requirement when training the model. As shown in Section 4.2, the memory requirement is reduced by over 40\%, while all the other baselines cannot reduce the memory requirement.
>
> 2. We attain much higher robustness than other baselines when the proportion of resource-sufficient clients is small, as shown in Section 4.3. Notice that "FedDynAT (100\% AT)" is actually not feasible in our resource-constrained settings since it requires all clients to have sufficient resources. Thus, it only acts as an upper bound in our experiments, and FADE approaches this upper bound with only a small proportion of resource-sufficient clients.
>
> 3. We attain better overall performance on clean and adversarial samples by adjusting the hyperparameter $\lambda_m$, as shown in Section 4.4. Notice that we are not trying to eliminate the trade-off between accuracy and robustness because it is one characteristic of AT. What we can achieve is a better balance point between them.
>
> Q3. The experiments do not reflect the degree of resource-constrained edge devices.
>
> A3. The degree of the resource-constrained edge devices is reflected by the model partitions used by different clients, as well as the ratio between resource-sufficient clients and resource-insufficient clients. We show the results of mixing 1-module clients, 2-module clients and 3-module clients in a fixed ratio in Tables 2 and 3. And we also vary the proportion of the 1-module (resource-sufficient) clients from 0 to 1 and show the corresponding robustness in Figure 4.
>
> Q4. "Large-Scale" is not reflected in the experiments.
>
> A4. "Large-Scale" in our paper mainly refers to a large number of clients and a large model. While the heterogeneity is enlarged with increasing numbers of clients, we adopt a setting with 100 clients which are more than 50 clients in FedRBN and 10 clients in FedDynAT. At the same time, a larger model increases the memory requirement on edge devices, and we use a 7-layer CNN in FMNIST and an 11-layer VGG in CIFAR-10, which are larger than the models used in FedDynAT and FedRBN. For the choice of datasets, using FMNIST and CIFAR-10 is following the most common manners in the other FL literature (Such as FedNOVA, FedDynAT, and FedRBN).
>
> Q5. The ablation study of the algorithm is insufficient.
>
> A5. FADE only has three components that can be adjusted:
>
> 1. The way we partition the model on different clients, i.e., the number of modules on each client and the ratio between clients with different numbers of modules. We show the results of different numbers of modules and different ratios of resource-sufficient clients in Section 4.3, which highlights the effectiveness of FADE with flexible model partitions.
>
> 2. The weight decay hyperparameter $\lambda$. We give a comprehensive discussion on it in Section 4.4, and the results show that the auxiliary weight decay can help us achieve better overall performance on clean and adversarial samples.
>
> 3. The feature perturbation $\epsilon_m$ for AT in each module. The influence of this parameter is simple: Increasing it will improve the robustness but decrease the clean accuracy. Therefore we only perform a linear search to find the optimal value of $\epsilon_m$ that leads to the best overall performance on clean and adversarial samples. Please see Appendix C for more details.
>
> Q6. The relationship between the theory and the resource-constrained setting.
>
> A6. Theorem 1 guarantees the convergence of Federated Decoupled Learning with different model partitions on different clients. This theoretically allows us to assign each resource-constrained device a model partition that fits in their resource budgets respectively.

---

> ### Comment · Reviewer_xYLU · 2022-11-22
> **Response to The Authors**
>
> Thanks to the authors for the further clarification and explanation. After reading the response and the new version of the paper, I think the technical significance of FADE than DGL is well highlighted. Considering the overall quality of the submission with those further clarifications, I would like to raise the score accordingly.

---

### Decision · Program_Chairs · 2023-01-20

**Decision:**

Reject

**Justification For Why Not Higher Score:**

See above

**Justification For Why Not Lower Score:**

NA

**Metareview: Summary, Strengths And Weaknesses:**

The paper proposes a framework named Federated Adversarial Decoupled Learning (FADE) to enable adversarial training on resource-constrained edge devices.  The authors provide the theoretical guarantee for the convergence of the proposed algorithm and conduct experiments to empirically verify the effectiveness of FADE w.r.t. to adversarial robustness.

Despite the fact that the reviewers raised their scores, they remained concerned about several issues. As a result, they maintained a "borderline" recommendation overall. Issues that remained were:
(a) The technical novelty of the proposed method is limited as it simply applies decoupled learning in the federated adversarial learning setting.
(b)  Experiments could be further improved, along the line indicated in the reviews.
(c) Forward passing is still required in the first training round, which impacts the efficiency of the proposed method.
(d) Presentation overall could be improved, along the suggestions provided in the last review

**Summary Of Ac-Reviewer Meeting:**

Reviewers appreciated the responses but were not very enthusiastic about the paper. Concerns are captured above.